# Low mutation rate in epaulette sharks is consistent with a slow rate of evolution in sharks

Ashley T. Sendell-Price [1,2,15], Frank J. Tulenko [3,15], Mats Pettersson [1], Du Kang [4], Margo Montandon [3], Sylke Winkler [5], Kathleen Kulb[5], Gavin P. Naylor[6], Adam Phillippy [7], Olivier Fedrigo[8], Jacquelyn Mountcastle[9], Jennifer R. Balacco[9], Amalia Dutra[10], Rebecca E. Dale[3], Bettina Haase[8], Erich D. Jarvis [8], Gene Myers [5,11], Shawn M. Burgess [7,16] ✉, Peter D. Currie [3,12,16] ✉, Leif Andersson [1,13,16] ✉ & Manfred Schartl [14,16] ✉

Sharks occupy diverse ecological niches and play critical roles in marine ecosystems, often acting as apex predators. They are considered a slow-evolving lineage and have been suggested to exhibit exceptionally low cancer rates. These two features could be explained by a low nuclear mutation rate. Here, we provide a direct estimate of the nuclear mutation rate in the epaulette shark (*Hemiscyllium ocellatum*). We generate a high-quality reference genome, and resequence the whole genomes of parents and nine offspring to detect de novo mutations. Using stringent criteria, we estimate a mutation rate of $7 \times 10^{-10}$ per base pair, per generation. This represents one of the lowest directly estimated mutation rates for any vertebrate clade, indicating that this basal vertebrate group is indeed a slowly evolving lineage whose ability to restore genetic diversity following a sustained population bottleneck may be hampered by a low mutation rate.

Sharks are members of one of the most basal of vertebrate clades, the Chondrichthyans, that emerged from mass extinction events in the Permian and Jurassic periods to radiate and dominate many marine food webs[1,2]. Modern sharks play important functional roles in the regulation and maintenance of a diverse range of marine ecosystems[3–5]. However, little is known about the evolutionary rate and adaptive potential of shark populations, a fact that has come into sharper focus with the emergence of the dual ecological pressures of overfishing and habitat loss. Specific drivers of overfishing in shark populations are particularly impacting. Firstly, shark populations are

[1]Department of Medical Biochemistry and Microbiology, Uppsala University, SE75123 Uppsala, Sweden. [2]Bioinformatics Research Technology Platform, University of Warwick, Coventry, UK. [3]Australian Regenerative Medicine Institute, Monash University, Victoria 3800, Australia. [4]The Xiphophorus Genetic Stock Center, Department of Chemistry and Biochemistry, Texas State University, San Marcos, TX 78666, USA. [5]Max-Planck Institute of Molecular Cell Biology and Genetics, 01307 Dresden, Germany. [6]Florida Museum of Natural History, University of Florida, Gainesville, FL 32611, USA. [7]Translational and Functional Genomics Branch, National Human Genome Research Institute, National Institutes of Health Bethesda, Bethesda, MD 20892, USA. [8]Vertebrate Genome Laboratory, Rockefeller University, New York, NY 10065, USA. [9]Research Center for Genomic and Computational Biology, Duke University, Durham, NC 27708, USA. [10]Cytogenetics and Microscopy Core, National Human Genome Research Institute, National Institutes of Health Bethesda, Bethesda, MD 20892, USA. [11]Center of Systems Biology Dresden, 01307 Dresden, Germany. [12]EMBL Australia, Victorian Node, Monash University, Clayton, Victoria 3800, Australia. [13]Department of Veterinary Integrative Biosciences, Texas A&M University, College Station TX77483, USA. [14]Developmental Biochemistry, Theodor-Boveri Institute, Biocenter, University of Würzburg, 97074 Würzburg, Germany. [15]These authors contributed equally: Ashley T. Sendell-Price, Frank J. Tulenko. [16]These authors jointly supervised this work: Shawn M. Burgess, Peter D. Currie, Leif Andersson, Manfred Schartl. ✉e-mail: burgess@mail.nih.gov; peter.currie@monash.edu; leif.andersson@imbim.uu.se; phch1@biozentrum.uni-wuerzburg.de

severely adversely affected by their incidental capture in fisheries directed at other species, with sharks caught as bycatch in the high seas pelagic longline fisheries being particularly impactful[6]. Secondly, many species are directly targeted by the 'fin trade', where shark fins are harvested for human consumption. This removes between 26 and 73 million sharks each year, with more than half of the species being under threat of extinction[7]. Thirdly, for many years specific shark populations have also been harvested by an additional, particularly pernicious, industry which produces shark cartilage extracts as dietary supplements for cancer prevention or treatment. The use of this product is based on the claim that sharks do not get cancer[8,9]. The shark cartilage supplement industry persists despite the clinical efficacy of shark cartilage-based treatments of cancer being directly refuted by clinical trials[10]. Furthermore, the existence of numerous studies documenting that different types of neoplasms do, in fact, occur in sharks has also failed to halt the use of shark cartilage supplements[11].

Exacerbating the intense fishing pressures currently facing shark populations is the extreme nature of the life history characteristics that are exhibited by most shark species. Extant sharks are slow-growing, reach sexual maturity late, and have few offspring. They exhibit some of the longest gestation periods and the highest levels of maternal investment in the animal kingdom[12]. This generally results in slow population growth and delayed recovery after population collapse. They are, therefore, particularly sensitive to unsustainable fishing practices and rapid changes in habitats[13–16]. How rapidly shark populations are able to evolve to counteract the mounting ecological threats that face them and rebound from historically low population densities will ultimately be dependent on the genetic diversity within populations, a value that itself is dependent on the germline mutation rate.

Mutations are the fundamental substrates of evolution because they generate variability within populations, enabling evolutionary change. The mutation rate (μ) is a crucial parameter for many calculations and predictive modelling in the fields of ecology and evolution, genetics, and genomics. Despite its importance, experimental determination of mutation rates in vertebrates has been strongly mammal-focused (Supplementary Table 1), including a recent study reporting

mutation rates in 68 vertebrate species[17]. Synonymous substitution rates for chondrichthyans have been reported to be lower than those of osteichthyans, suggesting a low intrinsic mutation rate[18]. In addition, a mitochondrial DNA sequence-based study[19] previously indicated that sharks might be a "slow molecular clock lineage", which—if true—would have consequences for our understanding of the evolution, ecology, and genomics of this basal vertebrate group.

Here we provide a direct estimate of the de novo mutation rate in a species of shark—*Hemiscyllium ocellatum* (the epaulette shark)—a small, benthic, oviparous species that inhabits coral reef environments in the waters north-east of Australia (Fig. 1a). The epaulette shark is the most studied member of the genus *Hemiscyllium* or "walking" sharks for which a recent comprehensive molecular phylogenetic analysis based on whole mitochondrial genome sequences of all nine currently recognised species has been completed[20]. Our development of captive breeding and pair mating protocols for the epaulette shark allows the development of this species as a general model system for shark research and allows us to genetically evaluate the mutation rate within a shark pedigree. To the best of our knowledge, our analysis defines the lowest directly estimated mutation rate for a vertebrate to date, indicating that this basal vertebrate group is a slowly evolving lineage. These results have the potential to at least partially explain the perception of a low rate of cancer in shark species, and they also illustrate an additional hurdle that sharks face as a clade in maintaining genetic diversity against an ever-increasing ratchet of ecological pressures.

## Results

### Development of husbandry and pedigree procedures for *Hemiscyllium ocellatum*

We developed infrastructure to house a captive broodstock of epaulette sharks (Fig. 1b). Reproductively mature adults were sourced from wild populations along the northeastern Australian Coast. Within this brood stock, we isolated a single captive female and male breeding pair. Epaulette sharks are oviparous, and females lay two to four eggs per month[21]. To avoid false paternity assignment due to possible sperm storage, the male and female sharks were maintained in isolation for a period of approximately 10 months prior to the onset of egg

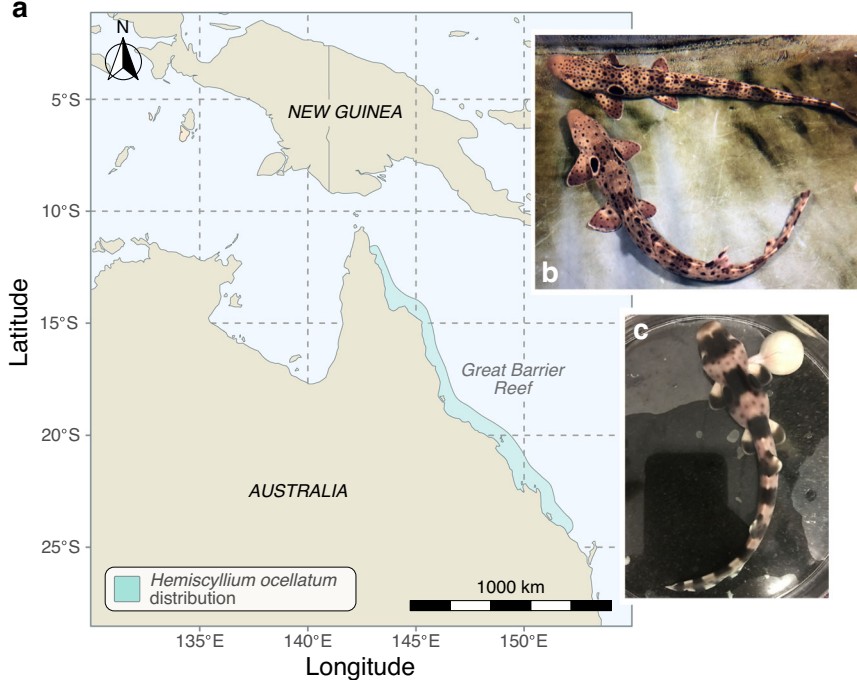

**Fig. 1 | Distribution and brood colony of epaulette sharks (*Hemiscyllium ocellatum*). a** Map showing epaulette shark geographic distribution according to refs. 20,50. **b** Isolated male and female adults used in this study. **c** Stage 37 pre-hatchling removed from egg case. A small remaining yolk ball is present on the right.

collection. Genomic DNA from ten F1 offspring was obtained from pre-hatching, whole embryos collected from the isolated breeding pair. Maternal and paternal genomic DNA were obtained from blood samples from each adult.

## High-quality assembly of the *Hemiscyllium ocellatum* genome

A trio from our pedigree comprising the male and the female and one of their progeny was used for the genome assembly. Following the phase 1 pipeline of the Vertebrate Genome Project[22], we used the "trio binning" strategy[23] to generate a haplotype-resolved genome assembly. For this method, we generated high coverage (113-135X) Illumina short-read sequences using genomic DNA isolated from maternal and paternal blood samples and 50X sequence coverage from a single F1 male offspring using PacBio Sequel II SMRT sequencing and genomic DNA isolated from tissue. A genome assembly was obtained using Canu[24]. Chromosome scaffolding was performed by integrating data from Hi-C (Dovetail) and optical mapping (Bionano) data. The strategy resulted in a paternal haplotype assembly (used as the reference assembly; GCA_020745735.1) of 3.98 Gb with 52 autosomes plus X (CM036711.1) and Y (CM036712.1) chromosomes. The assembly is of high quality, with 5667 contigs resolved into 26 large scaffolds and an

additional 1937 minor scaffolds. The N50 for the scaffolds is 83.6 Mb, and the scaffold L50 is 17 (Fig. 2, Table 1). A maternal haplotype assembly (GCA_020745765.1) was also assembled with a total length of 4.15 Gb. Karyotype analysis of cultured embryonic fibroblasts confirms the chromosome count from the genome assembly, demonstrating 52 autosome pairs and a pair of XY sex chromosomes (Fig. 3).

To annotate protein-coding genes, gene evidence from protein homology of other species, RNA-seq transcriptomes from epaulette shark embryos and ab initio predictions were integrated. A total of 18,225 protein-coding genes were annotated. The BUSCO completeness based on the vertebrata_odb9 data set was improved from 89.3% to 96.0% by the annotation process (Table 2). Totally, 1252 (6.8%) genes were annotated as pseudogenes. Of the 18,225 genes, 17,580 (96.5%) have a BLAST hit to the Swiss-Prot/RefSeq database. Of the protein-coding genes, 1275 (7%) are single exon genes. Additionally, 747 tRNA, 35 rRNA, 180 miRNA, and 854 other noncoding RNA genes were annotated (Table 2).

## Identification of de novo mutations

To provide a direct estimate of the de novo mutation rate in the epaulette shark, we generated 10X Genomics linked-reads sequencing

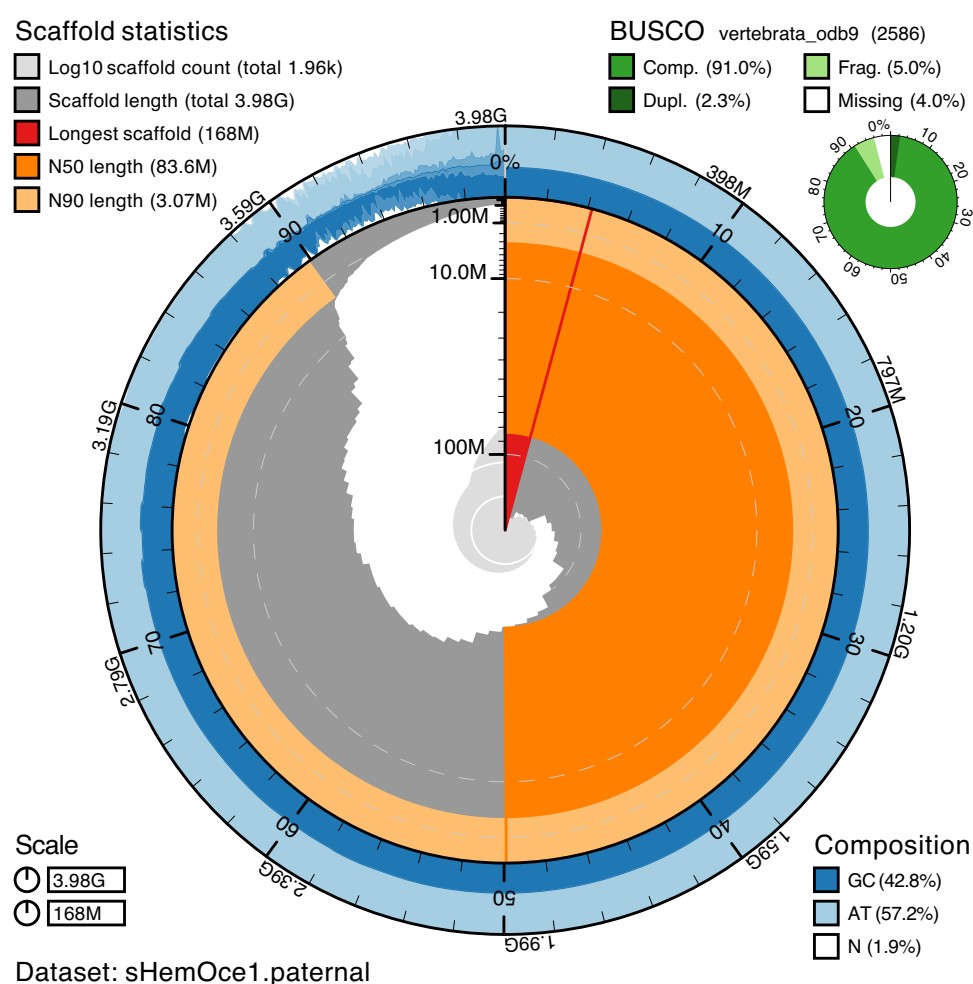

**Fig. 2 | Genome assembly metrics.** BlobToolKit76 snailplot showing N50 metrics and BUSCO gene completeness of the paternal genome assembly. The main plot is divided into 1000 size-ordered bins around the circumference, with each bin representing 0.1% of the assembly. The distribution of scaffold lengths is shown in dark grey, with the plot radius scaled to the longest scaffold present in the assembly (shown in red). Orange and pale-orange arcs show the N50 and N90 scaffold lengths, respectively. The pale grey spiral shows the cumulative scaffold count on a

log scale, with white scale lines showing successive orders of magnitude. The blue and pale-blue area around the outside of the plot shows the distribution of GC, AT and N percentages in the same bins as the inner plot. A summary of complete, fragmented, duplicated and missing BUSCO genes in the vertebrata_odb9 set is shown in the top right. See Supplementary Fig. 1 for snailplot of the maternal genome assembly.

data for nine F1 progeny produced during our captive breeding experiment. As identification of de novo mutations requires high sequence coverage, we sequenced each offspring to ~49–82× coverage (Supplementary Table 2). The resulting sequences, along with parental Illumina sequences (~113–135× coverage) previously generated for genome assembly construction, were aligned to our genome assembly and genotypes called at both variant and invariant sites using GATK[25] (see "Methods"). High genotype concordance confirmed a single paternity across the pedigree. In a known pedigree, de novo mutations can be identified as variant sites where an offspring carries an allele absent in both parents. However, offspring-parent genotype discordance can also arise via sequencing and alignment errors[26]. A standard genotype-calling pipeline (e.g., GATK best practices) will typically lead to most novel variants detected being false positives, as has been empirically demonstrated in the Atlantic herring[27]. Hence, prior to screening for candidate de novo mutations, we applied a strict genotype filtering pipeline[28] (Fig. 4, see Methods) designed to identify genomic positions where sample genotypes could be confidently called. This pipeline resulted in 333–457 Mb of sequence available per

trio for variant screening (Supplementary Table 2), which represents between 8.0% and 11.0% of the genome. Across these "callable" sites, we identified 12 candidate de novo mutations where offspring genotypes did not meet Mendelian expectations (Fig. 5A). Sanger sequencing/plasmid cloning confirmed four candidate de novo mutations as genuine and seven as false positives (Fig. 5A, Fig. 6), indicating a false positive rate of 63.6%. A similarly high false positive rate (61.4%) has been reported previously[29]. Consistent with germline mutations, all peak ratios for the two alleles of the confirmed de novo mutations were close to 1:1 (Fig. 5B). All four were transitions, in line with the general observation that transitions are more common than transversions[30]. Validation of the candidate de novo mutation on scaffold_28_mat (position: 17,411,576 bp) was not possible due to failed Sanger sequencing/difficulty cloning the target region in the focal offspring. Due to the presence of a flanking segregating SNP in the same sequencing reads, we could determine that the de novo mutation on scaffold_8_mat (position: 66,018,494 bp) was of paternal origin (Supplementary Fig. 2).

### Estimation of de novo mutation rate

To provide a correct estimate of the de novo mutation rate, we estimated the false negative rate. For a single offspring (ind1722), we simulated mutations at 947 invariant sites within high-confidence callable regions using the simulation tool SomatoSim[31]. We then repeated previously described genotype calling, variant filtering, and de novo mutation detection pipelines, compared genotype calls with expected genotypes based on the mutated sites and calculated the false negative rate. Our pipeline detected 910 out of 947 simulated mutations, indicating a low false negative rate of 3.9%. We took a conservative approach when estimating the mutation rate by including the candidate mutation on scaffold_28_mat in our calculations. This was the candidate mutation for which successful Sanger sequencing/plasmid cloning of the offspring carrying the mutation (ind2023) could not be conducted. We estimated the mutation rate per site per generation by dividing the number of de novo mutations identified by 2 x

### Table 1 | Statistics of the genome assembly

| | |
|---|---|
| Total sequence length | 3,983,483,121 |
| Total ungapped length | 3,905,918,519 |
| Gaps between scaffolds | 0 |
| Number of scaffolds | 1963 |
| Scaffold N50 | 83,580,160 |
| Scaffold L50 | 17 |
| Number of contigs | 5667 |
| Contig N50 | 8,845,575 |
| Contig L50 | 92 |
| Total number of chromosomes and plasmids | 55 |
| Number of component sequences (WGS or clone) | 1963 |

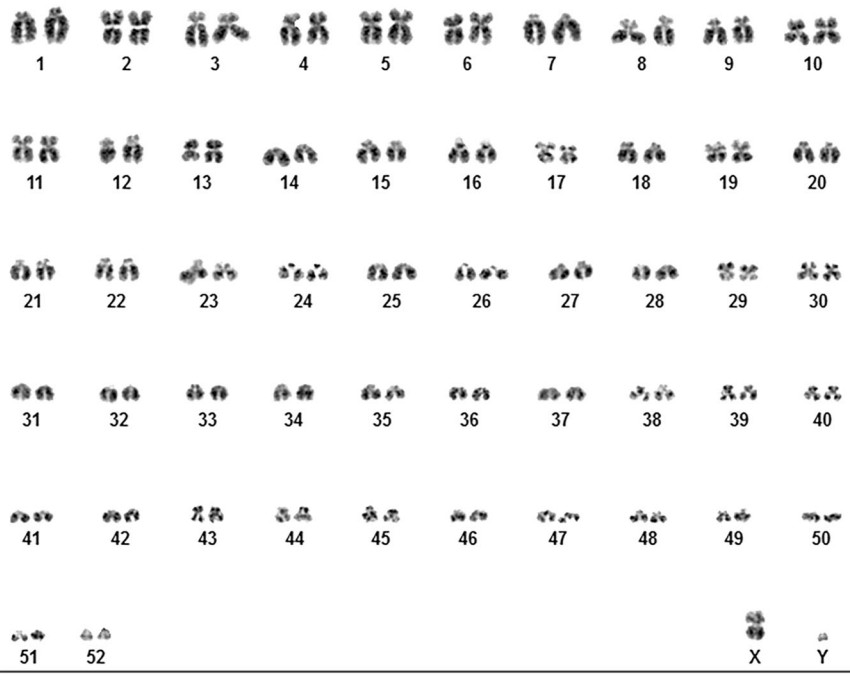

**Fig. 3 | Karyotype of the epaulette shark.** The epaulette shark karyotype has 52 pairs of autosomes plus X and Y sex chromosomes (53, XY). This includes six metacentric chromosomes (pairs: 2, 10, 17, 19, 29, X); 13 submetacentric chromosomes (pairs: 3, 4, 5, 6, 8, 11, 13, 30, 39, 40, 43, 44, 45); 24 acrocentric chromosomes (pairs: 1, 7, 9, 12, 15, 16, 18, 20, 21, 22, 23, 24, 26, 27, 33, 34, 35, 38, 46, 47, 48, 49, 51, Y); and 11 telocentric chromosomes (pairs: 14, 25, 28, 31, 32, 36, 37, 41, 42, 50, 52).

**Table 2 | Metrics of the genome annotation**

| | |
|---|---|
| Total_# of genes | 18,225 |
| Single_exon genes | 1275 (7.0%) |
| Multi_exon genes | 16,950 (93.0%) |
| Genes mapping_to_Pfam | 16,495 (90.5%) |
| Genes mapping_to_RefSeq/SwissProt | 17,580 (96.5%) |
| Genes with_RNA_support | 15,411 (84.6%) |
| Genes without_start_codon | 1090 (6.0%) |
| Pseudogenes_with_RNA | 732 (4.0%) |
| Pseudogenes_without_RNA | 520 (2.9%) |
| BUSCO analysis using the vertebrata_odb9 gene set ($n = 2586$) | |
| Before annotation | C:89.3% [S:86.8%, D:2.5%], F:4.4%, M:6.3% |
| After annotation | C:96.0% [S:92.0%, D:4.0%], F:2.2%, M:1.8% |

*C* complete, *S* single, *D* duplicate, *F* fragmented, *M* missing.

the total number of callable sites screened across the pedigree $(5/(2 \times 3,691,810,944) = 6.8 \times 10^{-10})$. By correcting for the estimated false negative rate, we obtain $7 \times 10^{-10}$ mutations per base pair per generation (95% CI: $1.4–14.1 \times 10^{-10}$, assuming that the mutations are Poisson distributed). Thus, a newborn epaulette shark carries approximately five single base de novo mutations compared with the corresponding estimate of 50–100 de novo mutations in newborn humans, despite the human genome being 25% smaller.

### Estimation of long-term effective population size

The relationship between nucleotide diversity ($\pi$), mutation rate ($\mu$) and effective population size ($N_e$) for diploid organisms is $\pi = 4N_e\mu$. Given this relationship, we calculated the long-term effective population size as $N_e = \pi/4\mu$. Using the nucleotide diversity observed in the two parents (mean $\pi = 0.002$, Fig. 7) and our estimated mutation rate of $7 \times 10^{-10}$ substitutions per site per generation, we obtained an estimated long-term $N_e$ of ~710,000 individuals.

## Discussion

Given the ecological and economic importance of chondrichthyans, and the relatively few genomic resources available from representatives of this clade[32], we sought to establish the epaulette shark as a laboratory model system and generate a high-quality, haplotype-resolved reference genome. We could then use this resource to estimate the de novo mutation rate for a shark species. The epaulette shark was chosen for this purpose because of the possibility of performing captive breeding and thereby ensuring a full-sib family for whole genome resequencing. Our finding of a de novo mutation rate of $7 \times 10^{-10}$ for the epaulette shark represents the lowest estimated rate yet reported for a vertebrate species, as illustrated in Fig. 8 and taking into account a recent study reporting mutation rates for 68 vertebrate species based on single trios[17]. Thus, our results indicate that sharks are a very slow molecular clock lineage.

The estimated mutation rate is 17-fold lower than in humans and an order of magnitude lower than the slowest evolving mammal recorded to date (Supplementary Table 1). However, it should be noted that this estimate reflects the mutation rate in the callable fraction of the genome, which does not include repeat regions (~44%). As replication of repetitive regions tends to be more error-prone, we acknowledge that the true genome-wide mutation rate is likely higher than reported here. However, the decreased ability to accurately call genotypes within repeat regions precludes unbiased screening within these regions. This issue is not unique to the epaulette shark, and as such similar caveats apply when estimating mutation rates in other species, meaning that results should be comparable across species.

**1. Whole genome resequencing**
- Conduct whole genome sequencing of experimental pedigree consisting of two wild-caught parents and nine captive-bred F1 offspring

**2. Read mapping & Genotype calling**
- Align sample reads against *H. ocellatum* genome assembly using BWA-mem
- Generate intermediate sample GVCFs using GATK HaplotypeCaller
- Combine intermediate sample GVCFs using GATK CombineGVCFs
- Conduct joint genotyping using GATK GenotypeGVCFs

**3. Apply hard-filters**
- Remove INDELs
- Remove repeat regions identified using RepeatMasker
- Remove regions with mappability <1
- Set genotypes to missing if GQ <20
- Remove sites with missing parental genotypes as these are not informative

**4. Define in-house filters**
- Remove sites with missing offspring genotypes
- Extract sites that are fixed for different alleles in parents and where all offspring are heterozygous - these represent high confidence heterozygous sites
- Extract quality parameters from high confidence heterozygous sites:
    - BaseQRankSum
    - ReadPosRankSum
    - MQ
    - QD
    - DP (per sample)
- For each quality parameter define lower and upper filtering bounds based on 5th & 95th percentiles

**5. Apply in-house filters**
- Apply in-house quality filters to hard filtered dataset
- Remove sites with missing parental genotypes
- Extract sites that are fixed for the same allele in both parents
- For each parent-offspring trio remove sites where focal offspring genotype is missing - remaining sites represent **"callable" informative sites**

**6. Detect candidate *de novo* mutations**
- For each trio identify putative mutations as any informative site where an offspring carries an allele absent in both parent

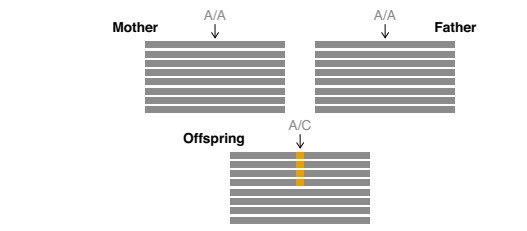

- Conduct secondary genotype calling of putative mutations using bcftools
- Retain putative mutations where GATK and bcftools genotypes match
- Remaining mutations are considered **high confidence candidate *mutations***

**Fig. 4 | In-house genotype filtering pipeline.** Genotype filtering and de novo mutation-calling pipelines were used in this study. Note: when identifying high-confidence heterozygous sites, genotype calls were considered homozygous in parents if the minor allele balance was <0.1 and heterozygous in offspring if the minor allele balance was ≥0.25. When applying in-house filters, we only applied the lower cut-off (5th percentile) for mapping quality (MQ) and quality by depth (QD) to prevent penalisation of high-quality sites.

There is a clear trend for lower mutation rates in poikilothermic vertebrates than in homoeothermic species because the three species with the lowest hitherto reported mutation rates are all fish (Fig. 8). As a correlation between nucleotide substitution rates and metabolism has been documented[33], a possible explanation for the low mutation rate is that the metabolic rate of sharks is up to ten times lower than in mammals of a similar size[34,35]. Given that epaulette sharks are restricted to warm tropical waters, even lower de novo mutation rates could be expected in shark species inhabiting cold waters where metabolic rates

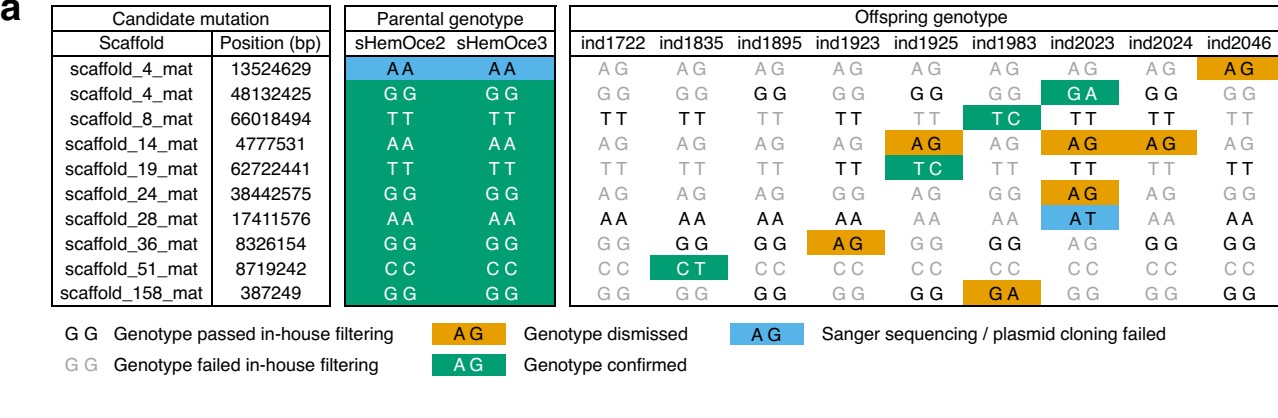

**Fig. 5 | Identification of candidate de novo mutations (DNMs). a** Sample genotypes at positions containing candidate de novo mutations. High-confidence genotypes are indicated by the black text, and low-confidence genotypes are indicated by the grey text. Genotypes were considered "high confidence" when GATK and bcftools derived genotypes matched. Parental genotypes and offspring candidate de novo mutations are coloured according to their validation status.
**b** Chromatograms showing parental and focal offspring Sanger sequences for confirmed de novo mutations.

are likely lower. For example, one of the longest-lived vertebrates is the Greenland shark (*Somniosus microcephalus*), which inhabits the most extreme latitudes of any shark species, and is exposed to some of the coldest water temperatures on the planet (as low as −1.8 °C)[36], and exhibits the lowest mass-specific metabolic rate reported for a shark[37].

Sharks do get cancers like all other vertebrates, although this has been suggested to occur at a lower rate than in other vertebrates[11]. As the shark skeleton is made of cartilage, the hypothesis was put forward that the high amount of cartilage in the shark body prevents the development of cancer. This was inferred from the well-established fact from mammalian cancer research that cartilage, including from

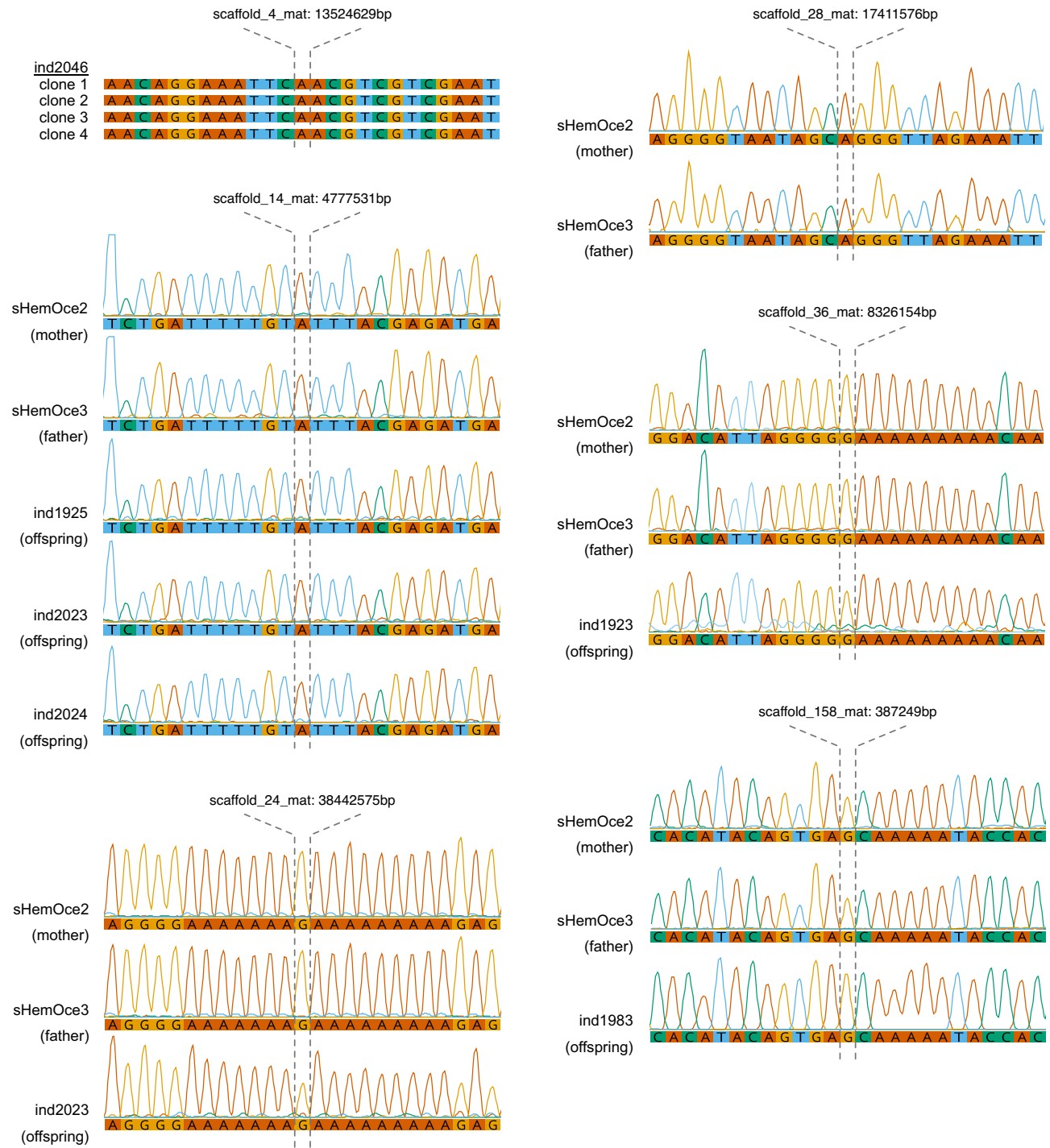

**Fig. 6 | False negative mutations.** Validation of parental and offspring sequences around false negative mutations via Sanger sequencing (scaffolds 14_mat, 24_mat, 28_mat, 36_mat, and 158_mat) or whole-plasmid Oxford Nanopore sequencing (scaffold_4_mat). Parental-only Sanger sequences are presented for the candidate mutation identified on scaffold_28_mat due to failed sequencing of the offspring carrying the candidate mutation.

sharks, inhibits neovascularization of tumours in vitro and thereby reduces their growth[38]. Indeed, anti-angiogenic factors have been isolated from mammalian and even shark cartilage[39]. The active principle behind this phenomenon is that biochemical components of cartilage can adsorb tumour-derived pro-angiogenic factors and thus inactivate them while others directly act as anti-angiogenesis molecules[10].

Following the first reports on anti-angiogenic activity from cartilage, the belief was nurtured that consuming shark cartilage as a "drug" could protect against cancer in humans. A whole industry developed which produces shark cartilage pills. Cartilage companies harvest over 100,000 sharks in US waters per month and up to tens of millions worldwide per year to create their products[40]. Shark cartilage, however, has not been shown to cure or modulate cancer progression in any way. It was ineffective in mouse tumour models[41]. This is also the conclusion from at least three randomised, FDA-approved clinical trials[42–44]. Most cancers originate from spontaneous or induced somatic mutations[45,46]. A study has shown that the somatic mutation rate is approximately one order of magnitude higher than the germline mutation rate[47]. Both values correlate, with a lower germline mutation

rate being accompanied by a lower somatic rate. Thus, we can also expect the somatic mutation rate in sharks to be low. Sharing their environments with other aquatic animals, which show a higher rate of neoplasms, we infer that the low spontaneous mutation rate of sharks could contribute to the low incidence of tumours suggested for sharks.

Based on the relationship between effective population size, nucleotide diversity, and the mutation rate, we estimated the long-term effective population size for the epaulette shark to be within the order of ~710,000 individuals. Such a large effective population size is unsurprising given that: (1) a previous mark-recapture census has estimated there to be thousands of epaulette sharks inhabiting the reefs surrounding Heron Island alone[48]; and (2) that the species has a broad geographic distribution[20,49,50]. While both a large population size and moderate nucleotide diversity (mean $\pi$ = 0.002) make the epaulette shark likely resilient to loss of diversity following short-term population perturbations, its ultra-low mutation rate means the species' ability to restore genetic diversity following a sustained population bottleneck would likely be low in particular if the bottleneck affects the entire species population.

Possible explanations for the low mutation rate in epaulette sharks are an intrinsic low mutation rate in poikilothermic species due to their low metabolic rate, combined with efficient purifying selection in a species with a large long-term effective population size that purges slightly deleterious mutations[51], for instance by selecting for efficiency in genes encoding the DNA repair machinery. In line with this suggestion, positive selection for genes involved in the maintenance of genome stability has previously been reported in elasmobranchs[52]. Extrapolating our findings to other shark species that lack the population size stability evident in epaulette sharks suggests a similar low mutation rate may result in long-term negative effects of population bottlenecks in already endangered and overfished species. Our study, therefore, provides compelling evidence for the need to prioritise preservation of the remaining genetic diversity of global shark populations.

## Methods
### Epaulette shark breeding and sampling
An adult epaulette shark brood stock was maintained in a closed, recirculating marine system, including three 5000 L tanks and a single 2100 L tank housed indoors in the Monash University Aquacore facility. Animals were originally purchased as sexually mature adults from Cairns Marine, which sourced wild-caught epaulettes from a collection area 100 nM south, 200 nM north and 150 nM East of Cairns (Queensland, Australia). Water temperature was maintained at approximately 25 °C, and a graded light cycle was used to mimic sunrise and sunset with a 12-h photoperiod. Sharks were fed a mixed diet that included glassies, pilchard, whiting, pipis and squid four times per week. Epaulette husbandry, breeding, and egg collection were carried out in accordance with approved Monash University Animal Ethics Project ID 30347, and blood samples from adult sharks were

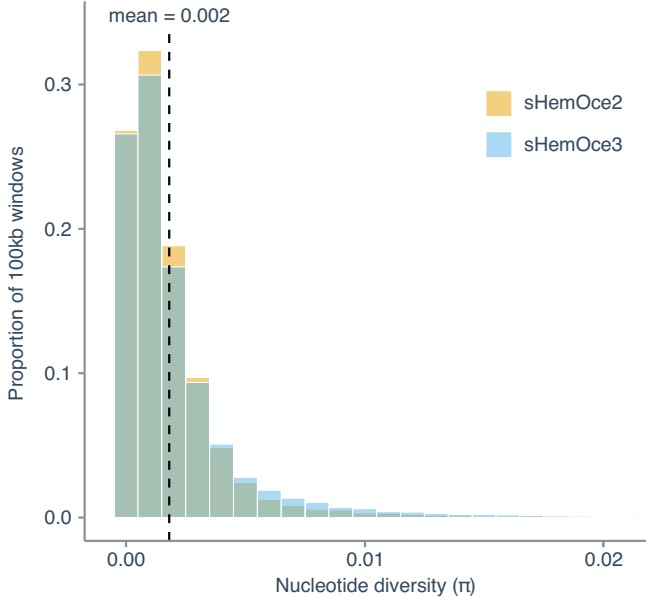

**Fig. 7 | Nucleotide diversity.** The distribution of nucleotide diversity ($\pi$) values was estimated in nonoverlapping 100 kb windows for parental samples (sHemOce2 and sHemOce3). The dashed line indicates the mean $\pi$ calculated across both samples. Source data are provided as a Source Data file.

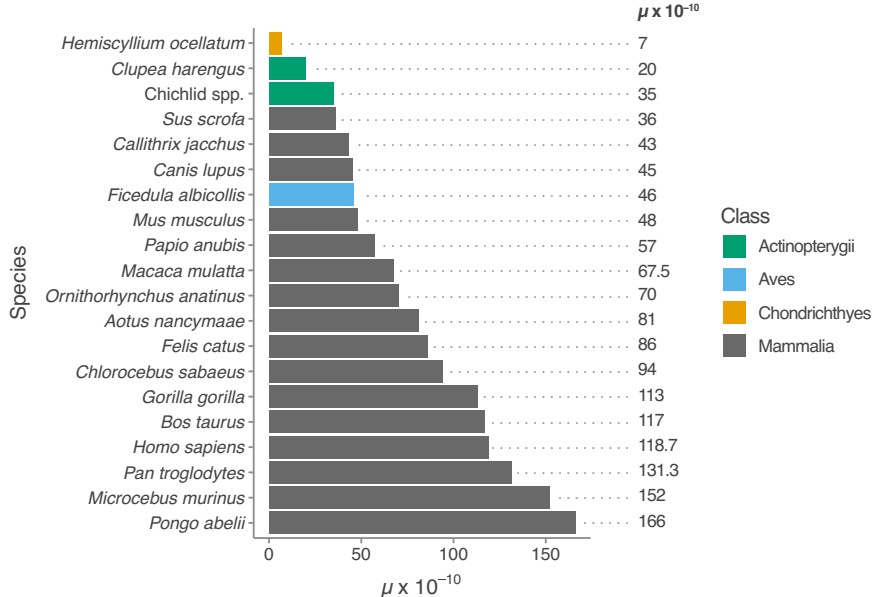

**Fig. 8 | Directly estimated vertebrate de novo mutation rates.** Where multiple estimates are available for a given species, see Supplementary Table 1, the mean is presented. Note: only mutation rate estimates that include validation of mutations via Sanger sequencing are reported.

collected according to approved Monash University Animal Ethics Project ID 13945. The adult breeding pair used for generating the trio assembly and pedigree analysis was housed in a custom-built 2100 L tank, which allowed the collection of eggs of known parentage. Newly laid eggs collected from the isolated breeding pair were tagged with their date of deposition, transferred to separate glass aquaria, and reared to late pre-hatching stages [Stages 37 and 38 according to refs. 53,54], flash frozen in liquid nitrogen and stored at −80 °C prior to DNA extraction. The adult tanks and egg-rearing aquaria were maintained on a common marine system and received the same seawater. For blood collection, adults were temporarily anaesthetised with Aqui-S and blood drawn from the caudal vein.

### Reference genome assembly and annotation

**Sampling.** Pre-hatchling sharks were flash frozen, and tissues were later dissected on dry ice. Whole blood from the parents was collected in EDTA-coated tubes and flash frozen and stored at −80 °C prior to DNA extraction.

**PacBio sequencing.** In total, 25 mg of spleen tissue was used to isolate genomic DNA for PacBio sequencing using the agarose plug Bionano Genomics protocol for cell culture DNA Isolation (#30026F). DNA quality was assessed by Pulsed Field Gel and quantified with a Qubit 2 Fluorometer. A total of 35.1 μg of "ultra" high molecular weight (uHMW) DNA was isolated. 12.4 μg of uHMW DNA was sheared using a 26 G blunt end needle (PacBio protocol PN 101-181-000 Version 05). A large-insert PacBio library was prepared using the Pacific Biosciences Express Template Prep Kit v2.0 (#100-938-900) following the manufacturer's protocol. The library was then size-selected (>20 kb) using the Sage Science BluePippin Size-Selection System. After size selection, we obtained 2.2 μg of the final library (55.2 ng/μl) with an average size of 54 kb. The PacBio Library was sequenced on 7 PacBio 8 M SMRT cells on the Sequel instrument with the Sequel® II Sequencing Plate 1.0 using the Sequel® II Binding Kit 1.0, capturing 15 h movies with no pre-extension time.

**Bionano measurements.** In total, 37 mg of heart tissue was used for isolating genomic DNA for PacBio using the agarose plug Bionano Genomics protocol for cell culture DNA Isolation (#30026 F). uHMW DNA quality was assessed by Pulsed Field Gel and quantified with a Qubit 2 Fluorometer. A total of 15.17 μg of uHMW DNA was isolated. uHMW DNA was labelled for Bionano Genomics optical mapping using the Bionano Prep Direct Label and Stain (DLS) Protocol (30206E) and run on one Saphyr instrument flow cell.

**Hi–C.** was performed by Arima using Arima v1 chemistry (restriction sites: GATC, GANTC) and sequenced on an Illumina NovaSeq 6000 S4 module using 150 bp paired-end (PE) chemistry (~60× coverage).

**10× Genomics linked-reads.** Unfragmented HMW DNA was used to generate a linked-reads library on the 10× Genomics Chromium platform (Genome Library Kit & Gel Bead Kit v2 PN-120258, Genome Chip Kit v2 PN-120257, i7 Multiplex Kit PN-120262). We sequenced this 10× library on an Illumina NovaSeq S4 module using 150 bp PE chemistry (~60× coverage).

**Parental Illumina short-read sequencing.** In total, 10 μl of whole blood was used for each parent, and DNA was isolated using the Qiagen QIAmp DNA Blood Kit (Cat. # 51104), DNA was ligated to Illumina adaptors using the Illumina DNA PCR-Free Prep Tagmentation (Cat. # 20027213). We sequenced this library on an Illumina NovaSeq 6000 S4 module using 150 bp PE chemistry (~60× coverage).

**Genome assembly.** The VGP pipeline v1.6 was used to assemble this genome[22]. We used TrioCanu v. 2.1[23] (https://canu.readthedocs.io/en/

latest/index.html) to assemble the PacBio contigs, and arrow/variantCaller v.2.3.3 was used to polish the contigs with PacBio data. The paternal and maternal assemblies were "purged" to remove false duplicates using purge_dups v.1.2.5[55] (https://github.com/dfguan/purge_dups). The two haplotypes were then scaffolded separately using scaff10x v.4.2 (https://github.com/wtsi-hpag/Scaff10X) for the 10X data, Bionano Solve v.3.6.1_11162020 for the optical maps and Salsa2 HiC v. 2.2[56] (https://github.com/marbl/SALSA) for the HiC data. Finally, three rounds of polishing were applied to the two assemblies simultaneously. First, the raw CLR PacBio reads were mapped using the PacBio version of minimap2[57] (https://github.com/PacificBiosciences/pbmm2) and then polished using variantCaller v.2.3.3. Two rounds of polishing 10× data were mapped to the two haplotypes using Longranger v.2.2.2 (https://github.com/10XGenomics/longranger) and polished using freebayes v.1.3[58] (https://github.com/freebayes/freebayes). Assemblies were evaluated using Merfin v1.0[59] (https://github.com/arangrhie/merfin). Assemblies were evaluated with Merqury[60]. The two final haplotype assemblies were curated following the curation process described in ref. 61. The mitochondrial genome was assembled using mitoVGP[62] (https://github.com/gf777/mitoVGP).

**Repeat masking.** For creating a repeat masked assembly (see https://genome.ucsc.edu) WindowMasker was run with the following parameters:

```
windowmasker -mk_counts true \
-input GCA_020745765.1_sHemOce1.mat.decon.unmasked.fa \
-output wm_counts windowmasker -ustat wm_counts -sdust
true \
-input GCA_020745765.1_sHemOce1.mat.decon.unmasked.fa \
-output windowmasker.intervals
perl -wpe 'if (s/^>lcl\|(.*)\n$//) {$chr = $1;} \if
(/^(\d+) - (\d+)/) {\$s=$1; $e=$2+1; s/(\d+) -
(\d+)/$chr\t$s\t$e/;}' windowmasker.intervals > window
masker.sdust.bed
```

The windowmasker.sdust.bed included masking for areas of the assembly that are gaps. The file was 'cleaned' to remove those areas of masking in gaps, leaving only the sequence masking. The final result covers 1,838,924,616 bases in the assembly size 4,149,461,884 for a percent coverage of 44.32%.

**Annotation.** Protein coding genes were annotated by collecting and synthesising the gene evidence from homologous alignment, RNA-seq mapping and ab initio prediction. A pipeline from our previous study[63] was used in this process. For homology evidence, 458,466 protein sequences were aligned to the assembly using Exonerate[64] (https://www.ebi.ac.uk/about/vertebrate-genomics/software/exonerate) and Genewise[65] (https://www.ebi.ac.uk/Tools/psa/genewise/) for gene structure determination, respectively. Those protein sequences were collected from the vertebrate database of Swiss-Prot (https://www.uniprot.org/statistics/Swiss-Prot), RefSeq database (https://www.ncbi.nlm.nih.gov/refseq/, proteins with ID starting with "NP" from "vertebrate_other") and the NCBI genome annotation of human (GCF_000001405.39_GRCh38, https://www.ncbi.nlm.nih.gov/datasets/genome/GCF_000001405.39/), zebrafish (GCF_000002035.6, https://www.ncbi.nlm.nih.gov/datasets/genome/GCF_000002035.6/), platyfish (GCF_002775205.1, https://ncbi.nlm.nih.gov/datasets/genome/GCF_002775205.1/), medaka (GCF_002234675.1, https://ncbi.nlm.nih.gov/datasets/genome/GCF_002234675.1/), elephant shark (GCF_018977255.1, https://ncbi.nlm.nih.gov/datasets/genome/GCF_018977255.1/), Asian bonytongue (GCF_900964775.1, https://www.ncbi.nlm.nih.gov/datasets/genome/GCF_900964775.1/), coelacanth (GCF_000225785.1, https://ncbi.nlm.nih.gov/datasets/genome/GCF_000225785.1/) and western clawed frog (GCF_000004195.4, https://ncbi.nlm.nih.gov/

datasets/genome/GCF_000004195.4/). For transcriptome evidence, RNA-seq reads from mixed tissue collected from stage 23 and 27 epaulette shark embryos were aligned on the assembly using HISAT[66] (http://daehwankimlab.github.io/hisat2/). The gene models were determined using StringTie[67] (https://ccb.jhu.edu/software/stringtie/), and in parallel aligned reads were assembled using Trinity[68]. The resulting transcripts were then aligned to the assembly to determine the gene structure using Splign[69] (https://www.ncbi.nlm.nih.gov/sutils/splign/splign.cgi). For ab initio prediction, AUGUSTUS[70] (https://bioinf.uni-greifswald.de/augustus/) was trained using those "good genes" that were determined consistently by Exonerate, Genewise, StringTie and Splign. The trained AUGUSTUS was then run for the ab initio gene prediction with all the gene models obtained above as hints. To synthesise this gene evidence into a final consistent set of annotations, we first clustered overlapped homology gene models and, for each cluster, kept the one best supported by transcriptome evidence. The terminal exons were replaced when they encountered a replacement that was better supported by transcriptome evidence. Genome regions with no homologous gene predicted by ab initio gene models were recruited when they were 100% supported by transcriptome evidence.

The final annotated gene set was blasted through databases of Pfam (https://pfam.xfam.org/), BUSCO[71] (https://busco.ezlab.org/), Swiss-Prot (https://www.uniprot.org) and RefSeq (https://www.ncbi.nlm.nih.gov/refseq/) to check for protein domains, assess annotation completeness and assign gene symbol and name. Genes that are heavily covered by repeat elements, with low homologous coverage, lack transcriptome evidence and/or show no similarity to Pfam/Swiss-prot/RefSeq database were judged as poor quality and were discarded from the final gene set.

### Karyotype analysis

Epaulette shark embryos were dissected from egg cases at Stages 32–33 and used to seed fibroblast culture as described[72], with minor modifications. To prevent contamination, embryos were soaked in povidone-iodine (Betadine) solution for ten seconds and washed in shark PBS containing 1% antibiotic–antimycotic solution (Thermo Fisher Scientific GIBCO). Tissue was then macerated, plated on 24 well plates coated with rat tail collagen I following manufacturer recommendations (Thermo Fisher Scientific-GIBCO) and cultured in LDF media. Cultures were incubated at 26 °C in a humidified atmosphere with 5% $CO_2$. After 1 week, primary cultured fibroblasts were subcultured using 1.46 U/ml Dipase II (Thermo Fisher Scientific-GIBCO) in PBS supplemented with 299 mM urea and 68 mM NaCl. At maximum proliferation, cells were treated with colcemid (150 ng/ml) for 1.5 to 3 h, harvested and treated with 0.075 M KCl for 40 min. Cells were subsequently fixed in methanol:acetic acid (3:1), and the cell suspension was dropped onto glass slides and air-dried for DAPI banding analysis.

### High coverage resequencing of offspring individuals

Muscle tissue was quickly cut from the frozen embryo (kept frozen on dry ice) by Dremel multifunctional tool Model 4000 with EZ SpeedClic Φ38mm at a speed 30,000 rpm and then stored at −80 °C until DNA extraction.

High molecular weight (HMW) genomic DNA (gDNA) was extracted from one tissue section using Nanobind Tissue Big DNA Kit (Pacific Biosciences of California, USA) according to the manufacturer's protocol [Nanobind Tissue Big DNA Kit Handbook v1.0 (11/2019) -Standard TissueRuptor II HMW Protocol]. The quantity of gDNA was estimated with Qubit (Qubit dsDNA BR assay Kit) and NanoDrop Spectrophotometer (Thermo Fisher Scientific, Waltham, MA, USA), while the integrity of the DNA was verified using pulse field gel electrophoresis with the Pippin Pulse™ device (SAGE Science).

**10x genomic linked read sequencing.** HMW gDNA was used for 10× genomic linked read sequencing following the manufacturer's instructions (10× genomics Chromium™ Reagent Kit v2, revision B). In brief, 1 ng of HMW gDNA was amplified in 10× genome in gel beads (Gel Bead-In-Emulsions = GEM), making use of the Chromium™ device. Individual gDNA molecules were amplified in these individual GEMS in an isothermal incubation using primers that contain a specific 16 bp 10× barcode and the Illumima® R1 sequence. After breaking the emulsions, pooled amplified barcoded fragments were purified, enriched and went into Illumina sequencing library preparation as described in the protocol. Sequencing was done on a NovaSeq 6000 S1 flow cell using the 2 × 150 cycles paired-end regime plus 8 cycles of i7 index.

### Detection of candidate de novo mutations

**Read mapping and variant calling.** Offspring and parental Illumina sequences were aligned to the maternal haplotype genome assembly using BWA-mem (https://github.com/lh3/bwa) v0.7.17[73] (https://github.com/lh3/bwa). Sequence alignments were used to call variants via the GATK[25] (https://gatk.broadinstitute.org/) v4.2.0 *HaplotypeCaller*, which performs simultaneous calling of SNPs and Indels via local de novo assembly of haplotypes (see GATK manual for details). We ran *HaplotypeCaller* separately for each individual to generate intermediate genomic VCF files (gVCF). Following this, we used the *CombineGVCFs* and *GenotypeGVCFs* modules in GATK to merge gVCF records from each individual using the multi-sample joint aggregation step that combines all records, generates correct genotype likelihoods, re-genotypes the newly merged records and reannotates each of the called variants[25]. Raw variant calls were then filtered using GATK SelectVariants to retain only monomorphic sites and biallelic Single Nucleotide Polymorphisms (SNPs) for downstream analyses.

**Genotype filtering.** We excluded genotype calls from repetitive regions detected using Repeat Masker (https://github.com/rmhubley/RepeatMasker) v4.1.0 and genomic regions with a mappability score <1. Mappability was calculated using GENMAP[74] (https://github.com/cpockrandt/genmap) v1.3.0 using a k-mer length of 100 and a maximum of two mismatches. Second, we excluded any genotype call with a genotype quality (GQ) score <20 on the basis that genotype accuracy rapidly declines below this threshold (see GATK manual). Further, we removed sites where either parental genotype was missing, as these are not informative. Following this, we extracted a subset of sites where parents were homozygous for different alleles and all nine offspring were heterozygous (genotype calls were considered homozygous in parents if the minor allele balance was <0.1 and heterozygous in offspring if the minor allele balance was ≥0.25). From these high-confidence heterozygous sites, we extracted the following quality annotations from the VCF INFO field: base quality rank sum; read position rank sum; mapping quality; and quality by depth. In addition, from the VCF FORMAT field, we extracted the depth of coverage annotations for individual genotypes. We then examined their distributions in the high-confidence heterozygous sites and used the 5th and 95th percentiles calculated for each quality annotation as standard cut-offs to filter biallelic and monomorphic sites in our entire dataset. Note: for mapping quality and quality by depth, we only applied the lower cut-off (5th percentile) to prevent penalisation of high-quality sites. Sites that passed our in-house filtering pipeline were considered high-confidence "callable" sites.

**De novo mutation calling.** From the filtered dataset generated in the previous step, we identified candidate de novo mutations as sites where both parents were homozygous for the same allele and at least one offspring carried a variant allele in the heterozygous state. We conducted secondary genotype calling at these positions, using the

*mpileup* and *call* functions of bcftools[75] v1.14 (https://samtools.github.io/bcftools/), and considered sites as true candidates when GATK and bcftools genotypes matched, and when the putative mutation was supported by at least 25% reads, i.e. had a minor allele balance ≥0.25.

**Estimation of the false negative rate.** We simulated mutations by introducing variants directly into sample BAM files using the Single Nucleotide Variant (SNV) simulation tool SomatoSim v1.0.0[31] (https://github.com/BieseckerLab/SomatoSim). The advantages of this approach compared to generating synthetic reads from a reference file is that this approach allows for error profiles to be preserved and does not limit variant allele frequencies (VAFs), variant locations, or the number of variants that can be simulated. For a single offspring (ind1722), we simulated mutations at 947 invariant sites within high-confidence callable regions. Each mutated site had its frequency of mutated reads determined by sampling from the observed frequency distribution of callable heterozygous sites in the original dataset. We then repeated previously described genotype calling, genotype filtering and de novo mutation detection pipelines, compared the SNP calls with expected genotypes based on the mutated sites and calculated the false negative rate.

**Experimental validation and parental origin of de novo mutations**

To confirm the authenticity of candidate de novo mutations, we performed Sanger sequencing of the genomic regions around each candidate in both parents and all nine offspring. To confirm the sequence of the parents at candidate mutation sites, genomic DNA was extracted from blood samples using a PureLink™ Genomic DNA Mini Kit (Thermofisher) and used as a template for PCR amplification. PCR was performed using Phusion® High Fidelity DNA Polymerase (NEB) or PrimeSTAR GXL DNA Polymerase (Takara). Primer pairs are summarised in Supplementary Table 3. Sanger sequencing was performed on amplified fragments with the respective forward and reverse PCR primers, or fragments were cloned into pGEM®Teasy vector (Promega) and sequenced from M13 forward and M13 reverse sites.

For mutations detected in offspring individuals, the same DNA samples used for whole genome sequencing were used. PCR and Sanger sequence-based screen PCR amplification of the region of interest was performed in a total volume of 10 μl making use of the Phusion Flash Mastermix (Thermo Scientific) with 2 μl input of genomic DNA, 0.5% DMSO and 0.5 μM forward and reverse primer. All details on primer sequences (target-seq-primer) and on PCR conditions are listed in Supplementary Table 3. Sanger sequencing was performed either with the respective forward and reverse PCR primers or, if required, with internally located dedicated sequencing primers to cover the region of interest. For offspring ind2046 the target region around the candidate mutation located on scaffold_4_mat (position 13524629) was cloned into pGEM®Teasy vector, and four single plasmid clones sequenced by Plasmidsaurus (www.plasmidsaurus.com), using Oxford Nanopore Technology MinION with coverage that exceeded 200x.

**Parental origin of de novo mutations**

We attempted to infer the parental offspring of verified de novo mutations based on the occurrence of flanking SNP alleles that segregated between the two parents (i.e., positions where parents were homozygous for different alleles) and occurred within the same Illumina read or mate-pair read. Due to the limited presence of segregating sites, this was only possible for a single de novo mutation.

**Estimation of nucleotide diversity and effective population size**

For parental samples, we estimated nucleotide diversity ($\pi$) in non-overlapping 100 kb windows using pixy[76]. Given the relationship between nucleotide diversity ($\pi$), mutation rate ($\mu$) and effective population size ($N_e$) for diploid organisms is $\pi = 4N_e\mu$, we extrapolated the effective population size using the formula: $N_e = \pi/4\mu$.

**Reporting summary**

Further information on research design is available in the Nature Portfolio Reporting Summary linked to this article.

## Data availability

Data used for genome assembly construction are available at: https://genomeark.github.io/genomeark-all/Hemiscyllium_ocellatum.html. Additional raw Illumina sequencing reads used for the detection of candidate de novo mutations have been deposited at NCBI under the BioProject PRJNA900175. Genome assemblies (paternal and maternal haplotypes) are available from NCBI under GenBank accession numbers GCA_020745735.1 and GCA_020745765.1. Both haplotypes are also available through the UCSC genome browser gateway (https://genome.ucsc.edu/h/GCA_020745735.1 and https://genome.ucsc.edu/h/GCA_020745765.1). Source data are provided in this paper.

## Code availability

Custom code used for the detection of candidate de novo mutations is available on GitHub.

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

## Acknowledgements

We gratefully acknowledge the contribution of the Long Read Team of the Dresden Concept Genome Centre. We thank M. Hof for the information and discussion. This study was supported by the DFG Research Infrastructure NGS-CC as part of the Next Generation Sequencing Competence Network (project 423957469) and grants from the Deutsche Forschungsgemeinschaft (SCHA 408/15-1) as part of the DFG Sequencing call to M.S., Vetenskapsrådet (2017-02907) and Knut and Alice Wallenberg Foundation (K.A.W. 2016.0361) to L.A., Australian Research Council Discovery Grant DP220102970 to P.D.C. and F.T., Florida Museum of Natural History to G.P.N., and an NHMRC Fellowship GNT1136567 to P.D.C. This research was supported in part by the Intramural Research Programme of the National Human Genome Research Institute (ZIAHG200386-06). The authors would like to acknowledge the use of computing resources at Uppsala Multidisciplinary Centre for Advanced Computational Science (UPPMAX) in carrying out this work.

## Author contributions

L.A. and M.S. conceived and supervised the study; F.T., P.C. and R.E.D. provided the biological material; A.P., O.F., J.M., J.B., B.H., G.P.N., A.D., E.D.J. and S.B. generated the reference genome assembly, A.D. and M.M. prepared the karyotype; F.T. and P.C. provided transcriptome sequence; D.K. performed the annotation; S.W. and G.M. sequenced the offspring; A.T.S.P. and M.P. analysed the pedigree data and identified candidate mutations; S.W., F.T., K.K. and A.T.S.P. validated the mutations; L.A., P.C., S.B. and M.S. interpreted the data; ATSP drafted the paper; S.B., P.C. and L.A. revised the paper with input from other authors; all authors approved the paper prior to submission.

## Funding

## Competing interests

The authors declare no competing interest.
