## [Peer Review File · Nature Communications]

Low mutation rate in epaulette sharks is consistent with a slow rate of evolution in sharksREVIEWER COMMENTS

Reviewer #1 (Remarks to the Author):

This is a very nicely written and a really interesting piece of work: the manuscript is framed interestingly to deliver the main finding: surprisingly and intriguingly low estimate of de novo mutation rate in a shark species. This finding is further discussed sensibly, and attempts are made to pin-point explanations for the low mutation rate in sharks. This is without doubt a paper that will get people's attention. The conclusions follow from the results, and the analyses appear to be conducted rigorously. The sample size of two parents and 9 F1-generation offspring is not impressive, but probably sufficient (under the assumption that the two parents are representative of the entire species) IF the estimated mutation rate is statistically well supported. Unfortunately, no confidence intervals are provided for the μ - we believe it is critical to see these to be certain that the conclusions hold. Apart of this, we do not have any major criticism, but some suggestions and reflections that might be worth considering as follows.

L48-49 Agreed. However, what one might be left thinking is that what role mutation rate is likely to play in restoration of genetic variation after population size bottleneck? Even if we assume that population experiences serve bottleneck and loses much of its genetic variation, many other things apart from mutation rate can be more important for the recovery of the genetic variability. These include for instance longevity/age structure – long-lived species like sharks can buffer against loss of diversity better than short-lived species. Other important factor is metapopulation structure – immigrants from other subpopulations can help to restore the lost diversity much faster than any increase in mutation rate. In this perspective, one could argue that the argument put forth on these lines is in principle correct, but at the same time, not necessarily very relevant unless one makes lots of assumptions. Some readers might go even so far as to claim that this argument is a red herring.

L54. “adaptive potential” rather than “adaptive ability”?

L172. See above. There should be a way to estimate confidence intervals for this estimate.

L181. The callable genome (CG) size applied here seemed quite close to the total genome size (more than 3Gb) rather than what had been mentioned earlier ranging from 8% to 11% of the reference genome size. Could it be that this refers to a sum of CGs for all offspring that were used to calculate an overall μ for the entire pedigree in this paper? μ for a pedigree is normally calculated by averaging the per-offspring μ , rather than summing over them. Since the estimate of μ depends heavily on the size of CG, it is quite important that the authors disclose (perhaps in the supplementary materials) how exactly this key parameter was calculated. Also, we note that the CG ranged from 45% to 91.5% in the earlier studies included into the “Mutationathon” paper in eLife. Hence, it is a bit worrying to see very small CG estimates (8 to 11%) in this paper. Some discussion about this in the ms could be in place. How do the authors explain the small callable genome size despite the deep sequencing coverage of the samples?

L186. Perhaps make it clear that this is the long-term equilibrium N_e – not the current.

L203. Generalization from one shark species (two parents) to entire shark clade is quite a jump. In the same vein, isn't this particular species quite exceptional in its biology? It lives in shallow waters and can tolerate extreme hypoxia and even anoxia. Is it thinkable that these stressful conditions have selected for efficient DNA reparation machinery and the μ for this species might not be representative for the entire shark clade?

L246-255. See the earlier comment above: this seems bit naïve conclusion given that other factors than mutation rate are likely to be more important for restoring genetic variability. Also, it would be perhaps a good idea discuss the interpretation of the N_e estimate and its assumptions.

L260-262. Please provide reference.

L268. The parental fish were kept in captivity for a very long time. Is thinkable that the meioses where the mutations occurred took place during the captive period? Is it thinkable that conditions in aquarium (e.g. temperature) might have influenced the mutation rates?

Should more details be given on the rearing conditions?

L453. From what was described here, the CG had been worked out by applying the “in-house filtering pipeline” which was a series of filters to ensure that the parental homozygotes and offspring heterozygotes are both true. It was nice to see that the filtering bounds were set based on distributions. However, Mendelian violation should be one of the steps identifying de novo mutations (numerator) rather than defining CG (denominator). It is not quite clear if this is how the authors’ pipeline treated these.

L460. Normally there is also an upper bound for allelic balance (AB) filter preventing offspring genotypes to be homozygotes for the alternative allele (1/1).

Fig 6. We believe standard errors or confidence intervals should be shown in this figure. Without seeing them, there is no way to know what to think of the data in the figure.

In addition to all the comments above, we note that there are number of small technical glitches in the text (missing full-stops, etc).

Reviewer #2 (Remarks to the Author):

This is a very well written manuscript and I enjoyed reading it. I have made comments and suggestion directly on the Word .doc. One of my concerns with the paper is the estimation of the mutation rate--please see comments in the Word .doc.

[Editorial note: Please see next page.]

Pedigree-based estimate of *de novo* mutation rate in the epaulette shark defines the lowest known vertebrate mutation rate

Ashley T. Sendell-Price^{1,2,&} (orcid: 0000-0002-1227-8929), Frank J. Tulenko^{3,&}, Mats Pettersson¹, Kang Du⁴, Margo Montandon³, Sylke Winkler⁵ (orcid: 0000-0002-0915-3316), Kathleen Kulb⁵, Gavin Naylor⁶, Adam Phillippy⁷, Olivier Federico⁸, Jacquelyn Mountcastle⁹, Jennifer R. Balacco⁹, Amalia Dutra¹⁰, Rebecca Dale³, Bettina Haase⁸, Erich Jarvis⁸, Gene Myers^{5,11}, Shawn M. Burgess^{7*} (orcid: 0000-0003-1147-0596), Peter D. Currie^{3,12*} (orcid: 0000-0001-8874-8862), Leif Andersson^{1,13*} (orcid: 0000-0002-4085-6968), Manfred Scharl^{4,14*} (orcid: 0000-0001-9882-5948)

¹ Department of Medical Biochemistry and Microbiology, Uppsala University, SE75123 Uppsala, Sweden.

² Bioinformatics Research Technology Platform, University of Warwick, Coventry, UK.

³ Australian Regenerative Medicine Institute, Monash University, Victoria, 3800, Australia.

⁴ The Xiphophorus Genetic Stock Center, Department of Chemistry and Biochemistry, Texas State University, San Marcos, Texas, USA.

⁵ Max-Planck Institute of Molecular Cell Biology and Genetics, 01307 Dresden, Germany.

⁶ Florida Museum of Natural History, University of Florida, USA.

⁷ Translational and Functional Genomics Branch, National Human Genome Research Institute, National Institutes of Health Bethesda, MD 20892, USA.

⁸ Vertebrate Genome Laboratory, Rockefeller University, New York, USA.

⁹ Research Center for Genomic and Computational Biology, Duke University, Durham, USA.

¹⁰ Cytogenetics and Microscopy Core, National Human Genome Research Institute, National Institutes of Health Bethesda, MD 20892 USA.

¹¹ Center of Systems Biology Dresden, 01307 Dresden, Germany

¹² EMBL Australia, Victorian Node, Monash University, Clayton, Victoria 3800, Australia

¹³ Department of Veterinary Integrative Biosciences, Texas A&M University, College Station, TX77483, USA.

¹⁴ Developmental Biochemistry, Theodor-Boveri Institute, Biocenter, University of Würzburg, Germany

& These authors contributed equally to this work

* Equally contributing senior authors

Correspondence and requests for materials should be addressed to:

Manfred Scharl (phchl@biozentrum.uni-wuerzburg.de), Leif Andersson (leif.andersson@imbim.uu.se),

Peter Currie (peter.currie@monash.edu) or Shawn Burgess (burgess@mail.nih.gov)

Sharks occupy diverse ecological niches and play critical roles in marine ecosystems, often acting as apex predators. They are considered a slow-evolving lineage and have been suggested to exhibit exceptionally low cancer rates. These two features could be explained by a low nuclear mutation rate. Here, we provide the first direct estimate of the nuclear mutation rate in a shark – the epaulette shark (*Hemiscyllium ocellatum*). The experimental design included generation of a high-quality reference genome, and whole genome resequencing of parents and nine offspring to detect *de novo* mutations. Using stringent criteria, we estimate a mutation rate of 6.8×10^{-10} per base pair, per generation. This represents the lowest directly estimated mutation rate for any vertebrate clade, indicating that this basal vertebrate group is indeed a slowly evolving lineage whose ability to restore genetic diversity following a sustained population bottleneck may be hampered by a low mutation rate.

Deleted:

Sharks are members of one of the most basal of vertebrate clades, the Chondrichthyans, that emerged from mass extinction events in the Permian and Jurassic periods to radiate and dominate many marine food webs^{1,2}. Modern sharks play important functional roles in the regulation and maintenance of a diverse range of marine ecosystems³⁻⁵. However, little is known about the evolutionary rate and adaptive ability of shark populations, a fact that has come into sharper focus with the emergence of the dual ecological pressures of overfishing and habitat loss. A number of specific drivers of overfishing in shark populations are particularly impacting. Firstly, shark populations are severely adversely affected by their incidental capture in fisheries directed at other species, with sharks caught as bycatch in the high seas pelagic longline fisheries being particularly impactful⁶. Secondly, many species are directly targeted by the ‘fin trade’, where shark fins are harvested for human consumption. This removes between 26 and 73 million sharks each year, with more than half of the species being under threat of extinction⁷. Thirdly, for many years specific shark populations have also been harvested by an additional, particularly pernicious, industry which produces shark cartilage extracts as dietary supplements for cancer prevention or treatment. The use of this product is based on the claim that sharks do not get cancer^{8,9}. The shark cartilage supplement industry persists despite the clinical efficacy of shark cartilage-based treatments of cancer being directly refuted by clinical trials¹⁰. Furthermore, the existence of numerous studies documenting that different types of neoplasms do in fact occur in sharks has also failed to halt the use of shark cartilage supplements¹¹.

Deleted:

Exacerbating the intense fishing pressures currently facing shark populations is the extreme nature of the life history characteristics that are exhibited by most shark species. Extant sharks are slow growing, reach sexual maturity late, and have few offspring. They exhibit some of the longest gestation periods and some of the highest levels of maternal investment in the animal kingdom¹². This generally results in slow population growth and delayed recovery after population collapse. They are therefore particularly sensitive to unsustainable fishing practices and rapid changes in habitats^{13–16}. How rapidly shark populations are able to evolve to counteract the mounting ecological threats that face them and rebound from historically low population densities will ultimately be dependent on the genetic diversity within populations, a value that itself is dependent on the germline mutation rate.

Mutations are the fundamental substrates of evolution because they generate variability within populations, enabling evolutionary change. The mutation rate (μ) is a crucial parameter for many calculations and predictive modelling in the fields of ecology and evolution, genetics, and genomics. Despite its importance, experimental determination of mutation rates in vertebrates has been strongly mammal focused (Supplementary Table 1) including a recent study reporting mutation rates in 68 vertebrate species¹⁷. Synonymous substitution rates for chondrichthyans have been reported to be lower than those of osteichthyans, suggesting a low intrinsic mutation rate¹⁸. In addition, a mitochondrial DNA sequence-based study¹⁹ previously indicated that sharks might be a "slow molecular clock lineage", which – if true – would have consequences for our understanding of the evolution, ecology, and genomics of this basal vertebrate group.

Here we provide a direct estimate of the *de novo* mutation rate in a species of shark – *Hemiscyllium ocellatum* (the epaulette shark) – a small, benthic, oviparous species that inhabits coral reef environments in the waters north-east of Australia (Fig. 1a). The epaulette shark is the most studied member of the genus *Hemiscyllium* or “walking” sharks for which a recent comprehensive molecular phylogenetic analysis based on whole mitochondrial genome sequences of all nine currently recognised species has been completed²⁰. Our development of captive breeding and pair mating protocols for the epaulette shark allows development of this species as a general model system for shark research and allows us the first genetic evaluation of the mutation rate within a shark pedigree. Our analysis defines the lowest directly estimated mutation rate for a vertebrate to date, indicating that this basal vertebrate group is a slowly evolving lineage. These results have the potential to at least partially explain the perception of

Deleted: (ref

Deleted:)

a low rate of cancer in shark species and they also starkly illustrate an additional hurdle that sharks face as a clade in maintaining genetic diversity against an ever-increasing ratchet of ecological pressures.

RESULTS

Development of husbandry and pedigree procedures for *Hemiscyllium ocellatum*

We developed infrastructure to house a captive broodstock of epaulette sharks (Fig. 1b). Reproductively mature adults were sourced from wild populations along the north-eastern Australian Coast. Within this brood stock, we isolated a single captive female and male breeding pair. Epaulette sharks are oviparous and females lay two to four eggs per month²¹. To avoid false paternity assignment due to possible sperm storage, the male and female shark were maintained in isolation for a period of approximately ten months prior to the onset of egg collection. Genomic DNA from ten F1 offspring was obtained from pre-hatching, whole embryos collected from the isolated breeding pair. Maternal and paternal genomic DNA were obtained from blood samples from each adult.

High quality assembly of the *Hemiscyllium ocellatum* genome

A trio from our pedigree comprising the male and the female, and one of their progeny was used for the genome assembly. Following the phase 1 pipeline of the Vertebrate Genome Project²², we used the “trio binning” strategy²³ to generate a haplotype resolved genome assembly. For this method we generated high coverage (113-135X) Illumina short-read sequences using genomic DNA isolated from maternal and paternal blood samples, and 50X sequence coverage from a single F1 male offspring using PacBio Sequel II SMRT sequencing and genomic DNA isolated from tissue. A genome assembly was obtained using Canu²⁴. Chromosome scaffolding was performed by integrating data from Hi-C (Dovetail) and optical mapping (Bionano) data. The strategy resulted in a paternal haplotype assembly (used as the reference assembly; GCA_020745735.1) of 3.98 Gb with 52 autosomes plus X (CM036711.1) and Y (CM036712.1) chromosomes. The assembly is of high quality with 5,667 contigs resolved into 26 large scaffolds and an additional 1,937 minor scaffolds. The N50 for the scaffolds is 8.63 Mb, the scaffold L50 is 17 (Fig. 2, Extended Data Table 1A). A maternal haplotype assembly (GCA_020745765.1) was also assembled with a total length of 4.15 Gb. Karyotype analysis of cultured embryonic fibroblasts confirms the chromosome count from

Deleted: o

genome assembly, demonstrating 52 autosome pairs and a pair of XY sex chromosomes (Extended Data Fig. 1).

To annotate protein-coding genes, gene evidence from protein homology of other species, RNA-seq transcriptomes from Epaulette shark embryos and *ab initio* predictions were integrated. A total of 18,225 protein-coding genes were annotated. The BUSCO completeness based on the vertebrata_odb9 data set was improved from 89.3% to 96.0% by the annotation process (Extended Data Table 1B). 1,252 (6.8%) genes were annotated as pseudogenes. Of the 18,225 genes, 17,580 (96.5%) have a BLAST hit to the Swiss-Prot/RefSeq database. Of the protein-coding genes, 1,275 (7%) are single exon genes. Additionally, 747 tRNA, 35 rRNA, 180 miRNA, and 854 other noncoding RNA genes were annotated (Extended Data Table 1B).

Identification of *de novo* mutations

To provide a direct estimate of the *de novo* mutation rate in the epaulette shark, we generated 10X Genomics linked-reads sequencing data for nine F1 progeny produced during our captive breeding experiment. As identification of *de novo* mutations requires high sequence coverage, we sequenced each individual to ~49-82 x coverage (Supplementary Table 2). The resulting sequences, along with parental Illumina sequences previously generated for genome assembly construction, were aligned to our genome assembly and genotypes called at both variant and invariant sites using GATK²⁵ (see Methods). High genotype concordance confirmed a single paternity across the pedigree. In a known pedigree *de novo* mutations can be identified as variant sites where an offspring carries an allele absent in both parents. However, offspring-parent genotype discordance can also arise via sequencing and alignment errors²⁶. A standard genotype-calling pipeline (e.g., GATK best practices) will typically lead to most novel variants detected being false positives, as has been empirically demonstrated in the Atlantic herring²⁷. Hence, prior to screening for candidate *de novo* mutations, we applied a strict genotype filtering pipeline (Fig. 3, see Methods) designed to identify genomic positions where sample genotypes could be confidently called. This pipeline resulted in 333-457 Mb of sequence available per trio for variant screening (Supplementary Table 2), which represents between 8.0% and 11.0% of the genome. Across these “callable” sites we identified 12 candidate *de novo* mutations where offspring genotypes did not meet Mendelian expectations (Fig. 4A). Sanger sequencing confirmed four candidate *de novo* mutations as genuine and six as false positives (Fig. 4A, Extended Data Figs. 2-5), indicating a false positive rate of 60%. A similarly high false positive rate (61.4%) has been reported previously²⁸. Consistent with germline mutations, all peak ratios

Commented [A1]: Up to 96 in offspring and up to 135 for the sire

for the two alleles of the confirmed *de novo* mutations were close to 1:1 (Fig. 4B). All four were transitions, in line with the general observation that transitions are more common than transversions²⁹. Validation of the remaining two candidate *de novo* mutations was not possible due to failed Sanger sequencing, however based on our estimated false positive rate the expected number of true *de novo* mutations is around 4.8. Due to the presence of a flanking segregating SNP in the same sequencing reads, we could determine that the *de novo* mutation on scaffold_8_mat (position: 66,018,494bp) was of paternal origin (Supplementary Figure 2).

Estimation of *de novo* mutation rate

To provide a correct estimate of the *de novo* mutation rate we estimated the false negative rate. For a single offspring (ind1722), we simulated mutations at 947 invariant sites within high confidence callable regions, using the simulation tool SomatoSim³⁰. We then repeated previously described genotype calling, variant filtering, and *de novo* mutation detection pipelines, and compared genotype calls with expected genotypes based on the mutated sites and calculated the false negative rate. Our pipeline detected 910 out of 947 simulated mutations, indicating a low false negative rate of 3.9%. We estimated the mutation rate per site per generation by dividing the expected number of true *de novo* mutations by 2 x the total number of callable sites screened ($4.8 / (2 \times 3,691,810,944) = 6.5 \times 10^{-10}$). By correcting for the estimated false negative rate we obtain: 6.8×10^{-10} mutations per base pair per generation. Thus, a newborn epaulette shark carries approximately five single base *de novo* mutations compared with the corresponding estimate of 50-100 *de novo* mutations in newborn humans, despite the human genome being 25% smaller.

Estimation of effective population size

The relationship between nucleotide diversity (π), mutation rate (μ) and effective population size (N_e) for diploid organisms is $\pi = 4N_e\mu$. Given this relationship, we calculated the effective population size as $N_e = \pi/4\mu$. Using the nucleotide diversity observed in the two parents (mean $\pi = 0.002$, Fig. 5) and our estimated mutation rate of 6.8×10^{-10} substitutions per site per generation, we obtained an estimated N_e of ~735,000 individuals.

DISCUSSION

Given the ecological and economic importance of chondrichthyans, and the relatively few genomic resources available from representatives of this clade³¹, we sought to establish the

Commented [A2]: But to dismiss these two occurrences changes the mutation rate, and instead of $(4.8 / (2 \times 3,691,810,944) = 6.5 \times 10^{-10})$, we would have $(6.8 / (2 \times 3,691,810,944) = 9.21 \times 10^{-10})$.

Commented [A3]: The calculation of the mutation rate needs to be explained better. Why not use EVERY possible mutation (see below)?

epaulette shark as a laboratory model system and generate a high quality, haplotype-resolved reference genome, the first such genome of its type for chondrichthyans. We could then use this resource to estimate the *de novo* mutation rate for a shark species. The epaulette shark was chosen for this purpose because of the possibility to perform captive breeding and thereby ensure a full sib family for whole genome resequencing. Our finding of a *de novo*-mutation rate of 6.8×10^{-10} for the epaulette shark represents the lowest estimated rate yet reported for a vertebrate species as illustrated in Fig. 6 and taking into account a recent study reporting mutation rates for 68 vertebrate species based on single trios¹⁷. Thus, our results indicate that sharks are a very slow molecular clock lineage.

Deleted:

The estimated mutation rate is 17-fold lower than in humans and an order of magnitude lower than the slowest evolving mammal. However, it should be noted that this estimate reflects the mutation rate in the callable fraction of the genome, which does not include repeat regions (~44%). As replication of repetitive regions tends to be more error-prone we acknowledge that the true genome-wide mutation rate is likely higher than reported here. However, decreased ability to accurately call genotypes within repeat regions precludes unbiased screening within these regions. This issue is not unique to the epaulette shark, and as such similar caveats apply when estimating mutation rates in other species, meaning that results should be comparable across species. There is a clear trend for lower mutation rates in poikilothermic vertebrates than in homeothermic species because the three species with the lowest hitherto reported mutation rates are all fish (Fig. 6). As a correlation between nucleotide substitution rates and metabolism has been documented³², a possible explanation for the low mutation rate is that the metabolic rate of sharks is up to 10 times lower than in mammals of a similar size^{33,34}. Given that epaulette sharks are restricted to warm tropical waters, even lower *de novo*-mutation rates could be expected in shark species inhabiting cold waters where metabolic rates are likely lower. For example, one of the longest lived vertebrates is the Greenland shark (*Somniosus microcephalus*), which inhabits the most extreme latitudes of any shark species, is exposed to some of the coldest water temperatures on the planet (as low as -1.8 °C)³⁵, and exhibits the lowest mass-specific metabolic rate reported for a shark³⁶.

Formatted: Font: (Default) Times New Roman

Sharks do get cancers like all other vertebrates, although this has been suggested to occur at a lower rate than in other vertebrates¹¹. As the shark skeleton is made of cartilage, the hypothesis was put forward that the high amount of cartilage in the shark body prevents the development of cancer. This was inferred from the well-established fact from mammalian cancer research,

that cartilage, including from sharks, inhibits neovascularization of tumors *in vitro* and thereby reduces their growth³⁷. Indeed, anti-angiogenic factors have been isolated from mammalian and even shark cartilage³⁸. The active principle behind this phenomenon is that biochemical components of cartilage can adsorb tumor-derived pro-angiogenic factors and thus inactivate them while others directly act as anti-angiogenesis molecules¹⁰.

Following the first reports on an anti-angiogenic activity from cartilage the belief was nurtured that consuming shark cartilage as a “drug” could protect against cancer in humans. A whole industry developed which produces shark cartilage pills. Cartilage companies harvest over 100,000 sharks in US waters per month and up to tens of millions worldwide per year to create their products³⁹. Shark cartilage, however, has not been shown to cure or modulate cancer progression in any way. It was ineffective in mouse tumor models⁴⁰. This is also the conclusion from at least three randomized, FDA-approved clinical trials^{41–43}. Most cancers originate from spontaneous or induced somatic mutations^{44,45}. A study has shown that the somatic mutation rate is approximately one order of magnitude higher than the germline mutation rate⁴⁶. Both values correlate, with a lower germline mutation rate being accompanied by a lower somatic rate. Thus, we can also expect the somatic mutation rate in sharks to be low. Sharing their environments with other aquatic animals, which show a higher rate of neoplasms, we infer that the low spontaneous mutation rate of sharks could contribute to the low incidence of tumors suggested for sharks.

Based on the relationship between effective population size, nucleotide diversity, and the mutation rate, we estimated the effective population size for the epaulette shark to be within the order of ~735,000 individuals. Such a large effective population size is unsurprising given that: (1) a previous mark-recapture census has estimated there to be thousands of epaulette sharks inhabiting the reefs surrounding Heron Island alone⁴⁷; and (2) that the species has a broad geographic distribution^{20,48,49}. While both a large population size and moderate nucleotide diversity (mean $\pi = 0.002$) make the epaulette shark likely resilient to loss of diversity following short-term population perturbations, its ultra-low mutation rate means the species’ ability to restore genetic diversity following a sustained population bottleneck would likely be low.

Possible explanations for the exceptionally low mutation rate in epaulette sharks are an intrinsic low mutation rate in poikilothermic species due to their low metabolic rate, combined with efficient purifying selection in a species with a large long-term effective population size that

purges slightly deleterious mutations⁵⁰, for instance by selecting for efficiency in genes encoding the DNA repair machinery. In line with this suggestion, positive selection for genes involved in the maintenance of genome stability has previously been reported in elasmobranchs. Extrapolating our findings to other shark species that lack the population size stability evident in epaulette sharks suggests a similar low mutation rate may result in long-term negative effects of population bottlenecks in already endangered and overfished species. Our study, therefore, provides compelling evidence for the need to prioritise preservation of the remaining genetic diversity of global shark populations.

METHODS

Epaulette shark breeding and sampling

An adult Epaulette shark brood stock was maintained in a closed, recirculating marine system including three 5000L tanks and a single 2100L tank housed indoors in the Monash University Aquacore facility. Animals were originally purchased from Cairns Marine, who sourced wild caught Epaulettes from a collection area 100nM south, 200nM north and 150nM East of Cairns (Queensland, Australia). Epaulette husbandry, breeding, and egg collection were carried out in accordance with approved Monash University Animal Ethics Project ID 30347, and blood samples from adult sharks were collected according to approved Monash University Animal Ethics Project ID 13945. The adult breeding pair used for generating trio assembly and pedigree analysis was housed in a custom built 2100L tank, which allowed the collection of eggs of known parentage. Eggs collected from the isolated breeding pair were reared to late pre-hatching stages [Stage 37-38 according to refs ⁵¹⁻⁵²] in separate glass aquaria, flash frozen in liquid nitrogen, and stored at -80C prior to DNA extraction. For blood collection, adults were temporarily anaesthetized with AQUI-S and blood drawn from the caudal vein.

Reference genome assembly and annotation

Sampling. Pre-hatchling sharks were flash frozen and tissues were later dissected on dry ice. Whole blood from the parents was collected in EDTA-coated tubes and flash frozen, and stored at -80 °C prior to DNA extraction.

PacBio sequencing. 25mg of spleen tissue was used to isolate genomic DNA for PacBio sequencing using the agarose plug Bionano Genomics protocol for cell culture DNA Isolation

Formatted: Superscript

(#30026F). DNA quality was assessed by Pulsed Field Gel and quantified with a Qubit 2 Fluorometer. A total of 35.1µg of “ultra” high molecular weight (uHMW) DNA was isolated. 12.4µg of uHMW DNA was sheared using a 26G blunt end needle (PacBio protocol PN 101-181-000 Version 05). A large-insert PacBio library was prepared using the Pacific Biosciences Express Template Prep Kit v2.0 (#100-938-900) following the manufacturer’s protocol. The library was then size selected (>20kb) using the Sage Science BluePippin Size-Selection System. After size-selection, we obtained 2.2µg of final library (55.2ng/µl) with an average size of 54kb. The PacBio Library was sequenced on 7 PacBio 8M SMRT cells on the Sequel instrument with the Sequel® II Sequencing Plate 1.0 using the Sequel® II Binding Kit 1.0 capturing 15h movies with no pre-extension time.

Bionano measurements. 37 mg of heart tissue was used for isolating genomic DNA for PacBio using the agarose plug Bionano Genomics protocol for cell culture DNA Isolation (#30026F). uHMW DNA quality was assessed by Pulsed Field Gel and quantified with a Qubit 2 Fluorometer. A total of 15.17µg of uHMW DNA was isolated. uHMW DNA was labeled for Bionano Genomics optical mapping using the Bionano Prep Direct Label and Stain (DLS) Protocol (30206E) and run on one Saphyr instrument flow cell.

Hi-C was performed by Arima using Arima v1 chemistry (restriction sites: GATC, GATC) and sequenced on an Illumina NovaSeq 6000 S4 module using 150bp paired-end (PE) chemistry (~60X coverage).

10X Genomics linked-reads. Unfragmented uHMW DNA was used to generate a linked-reads library on the 10X Genomics Chromium platform (Genome Library Kit & Gel Bead Kit v2 PN-120258, Genome Chip Kit v2 PN-120257, i7 Multiplex Kit PN-120262). We sequenced this 10X library on an Illumina NovaSeq S4 module using 150bp PE chemistry (~60X coverage).

Parental Illumina short-read sequencing. 10µl of whole blood was used for each parent and DNA was isolated using the Qiagen QIAmp DNA Blood Kit (Cat. # 51104), DNA was ligated to Illumina adapters using the Illumina DNA PCR-Free Prep Tagmentation (Cat. # 20027213). We sequenced this library on an Illumina NovaSeq 6000 S4 module using 150bp PE chemistry (~60X coverage).

Genome assembly. The VGP pipeline 1.6 was used to assemble this genome²². We used TrioCanu v. 2.1²³ (<https://canu.readthedocs.io/en/latest/index.html>), to assemble the PacBio

Deleted:

contigs and arrow/variantCaller v.2.3.3 was used to polish the contigs with PacBio data. The paternal and maternal assemblies were “purged” to remove false duplicates using `purge_dups` v.1.2.5⁵³ (https://github.com/dfguan/purge_dups). The two haplotypes were then scaffolded separately using `scaff10x` v.4.2 (<https://github.com/wtsi-hpag/Scaff10X>) for the 10X data, Bionano Solve v.3.6.1_11162020 for the optical maps and Salsa2 HiC v. 2.2⁵⁴ (<https://github.com/marbl/SALSA>) for the HiC data. Finally, three rounds of polishing were applied to the two assemblies simultaneously. First, the raw CLR PacBio reads were mapped using the PacBio version of `minimap2`⁵⁵ (<https://github.com/PacificBiosciences/pbmm2>) and then polished using `variantCaller` v.2.3.3. Two rounds of polishing 10X data was mapped to the two haplotypes using `Longranger` v.2.2.2 (<https://github.com/10XGenomics/longranger>) and polished using `freebayes` v.1.3⁵⁶ (<https://github.com/freebayes/freebayes>). Assemblies were evaluated using `Merfin`⁵⁷ (<https://github.com/arangrhi/merfin>). Assemblies were evaluated with `Mercury`⁵⁸. The two final haplotype assemblies were curated following the curation process described in ref ⁵⁹. The mitochondrial genome was assembled using `mitoVGP`⁶⁰ (<https://github.com/gf777/mitoVGP>).

Repeat masking. For creating a repeat masked assembly (see <https://genome.ucsc.edu>) `WindowMasker` was run with the following parameters:

```

windowmasker -mk_counts true \
-input GCA_020745765.1_sHemOcel.mat.decon.unmasked.fa \
-output wm_counts windowmasker -ustat wm_counts -sdust true \
-input GCA_020745765.1_sHemOcel.mat.decon.unmasked.fa \
-output windowmasker.intervals

perl -wpe 'if (s/^>lcl\|(.*)\n$/)) { $chr = $1; } \if (/^\(d+) - \(d+)/) {
\$$s=$1; $e=$2+1; s/\(d+) - \(d+)/$chr\t$$s\t$$e/; }' windowmasker.intervals >
windowmasker.sdust.bed

```

The `windowmasker.sdust.bed` included masking for areas of the assembly that are gaps. The file was 'cleaned' to remove those areas of masking in gaps, leaving only the sequence masking. The final result covers 1,838,924,616 bases in the assembly size 4,149,461,884 for a percent coverage of 44.32%.

Annotation. Protein coding genes were annotated by collecting and synthesizing the gene evidence from homologous alignment, RNA-seq mapping and *ab initio* prediction. A pipeline from our previous study⁶¹ was used in this process. For homology evidence, 458,466 protein

Commented [A4]: Make sure font and formatting are consistent throughout

sequences were aligned to the assembly using Exonerate⁶² (<https://www.ebi.ac.uk/about/vertebrate-genomics/software/exonerate>) and Genewise⁶³ (<https://www.ebi.ac.uk/Tools/psa/genewise/>) for gene structure determination, respectively. Those protein sequences were collected from the vertebrate database of Swiss-Prot (<https://www.uniprot.org/statistics/Swiss-Prot>), RefSeq database (proteins with ID starting with “NP” from “vertebrate_other”) and the NCBI genome annotation of human (GCF_000001405.39_GRCh38), zebrafish (GCF_000002035.6), platyfish (GCF_002775205.1), medaka (GCF_002234675.1), elephant shark (GCF_018977255.1), Asian bonytongue (GCF_900964775.1), coelacanth (GCF_000225785.1) and western clawed frog (GCF_000004195.4). For transcriptome evidence, RNA-seq reads from mixed tissue collected from stage 23 and 27 epauvette shark embryos were aligned on the assembly using HISAT⁶⁴ (<http://daehwankimlab.github.io/hisat2/>). Then gene models were determined using StringTie⁶⁵ (<https://ccb.jhu.edu/software/stringtie/>), and in parallel aligned reads were assembled using Trinity⁶⁶. The resulting transcripts were then aligned to the assembly to determine the gene structure using Splign⁶⁷ (<https://www.ncbi.nlm.nih.gov/sutils/splign/splign.cgi>). For *ab initio* prediction, AUGUSTUS⁶⁸ (<https://bioinf.uni-greifswald.de/augustus/>) was trained using those “good genes” that were determined consistently by Exonerate, Genewise, StringTie and Splign. The trained AUGUSTUS was then run for the *ab initio* gene prediction with all the gene models obtained above as hints. To synthesize these gene evidences into a final consistent set of annotation, we first clustered overlapped homology gene models, and for each cluster kept the one best supported by transcriptome evidence. The terminal exons were replaced when they encountered a replacement that is better supported by transcriptome evidence. In genome regions with no homologous gene predicted, *ab initio* gene models were recruited when they were 100% supported by transcriptome evidence.

The final annotated gene set was blasted through databases of Pfam (<https://pfam.xfam.org/>), BUSCO⁶⁹ and Swiss-Prot and RefSeq (<https://www.ncbi.nlm.nih.gov/refseq/>) to check for protein domains, assess annotation completeness and assign gene symbol and name. Genes that are heavily covered by repeat elements, with low homologous coverage, lack of transcriptome evidence and/or show no similarity to Pfam/Swiss-prot/RefSeq database were judged as poor-quality and were discarded from the final gene set.

Karyotype Analysis

Epaulette shark embryos were dissected from egg cases at Stages 32-33, and used to seed fibroblast culture as described⁷⁰, with minor modifications. To prevent contamination, embryos were soaked in povidone-iodine (Betadine) solution for ten seconds and washed in shark PBS containing 1% antibiotic-antimycotic solution (Thermo Fisher Scientific GIBCO). Tissue was then macerated, plated on 24 well plates coated with rat tail collagen I following manufacturer recommendations (Thermo Fisher Scientific-GIBCO), and cultured in LDF media. Cultures were incubated at 26°C in a humidified atmosphere with 5% CO₂. After one week, primary cultured fibroblasts were subcultured using 1.46 U/ml Dipase II (Thermo Fisher Scientific-GIBCO) in PBS supplemented with 299 mM urea and 68 mM NaCl. At maximum proliferation, cells were treated with colcemid (150ng/ml) for 1.5 to 3 h, harvested and treated with 0.075 M KCl for 40 min. Cells were subsequently fixed in methanol:acetic acid (3:1), and the cell suspension dropped onto glass slides and air-dried for DAPI banding analysis.

High coverage resequencing of offspring individuals

The muscle tissue was quickly cut from the frozen embryo (kept frozen on dry ice) by Dremel multifunctional tool Model 4000 with EZ SpeedClic Φ38mm at speed 30,000 rpm and then stored at -80°C until DNA extraction.

High molecular weight (HMW) genomic DNA (gDNA) was extracted from one tissue section using Nanobind Tissue Big DNA Kit (Pacific Biosciences of California, USA) according to the manufacturer's protocol [Nanobind Tissue Big DNA Kit Handbook v1.0 (11/2019) - Standard TissueRuptor II HMW Protocol]. The quantity of gDNA was estimated with Qubit (Qubit dsDNA BR assay Kit) and NanoDrop Spectrophotometer (Thermo Fisher Scientific, Waltham, MA, USA), while the integrity of the DNA was verified using pulse field gel electrophoresis with the Pippin PulseTM device (SAGE Science).

10x genomic linked read sequencing: HMW gDNA was used for 10X genomic linked read sequencing following the manufacturer's instructions (10X genomics ChromiumTM Reagent Kit v2, revision B). In brief, 1 ng of HMW gDNA was amplified in 10X genome in gel beads (Gel Bead-In-Emulsions = GEM) making use of the ChromiumTM device. Individual gDNA molecules were amplified in these individual GEMS in an isothermal incubation using primers that contain a specific 16bp 10X barcode and the Illumina® R1 sequence. After breaking the emulsions, pooled amplified barcoded fragments were purified, enriched and went into Illumina sequencing library preparation as described in the protocol. Sequencing was done on

a NovaSeq 6000 S1 flow cell using the 2x 150 cycles paired-end regime plus 8 cycles of i7 index.

Detection of candidate *de novo* mutations

Read mapping and variant calling: Offspring and parental Illumina sequences were aligned to the maternal haplotype genome assembly using BWA-mem2⁷¹ (<https://github.com/bwa-mem2/bwa-mem2>). Sequence alignments were used to call variants via the GATK²⁵ (<https://gatk.broadinstitute.org/>) v4.2.0 *HaplotypeCaller*, which performs simultaneous calling of SNPs and Indels via local *de novo* assembly of haplotypes (see GATK manual for details). We ran *HaplotypeCaller* separately for each individual to generate intermediate genomic VCF files (gVCF). Following this, we used the *CombineGVCFs* and *GenotypeGVCFs* modules in GATK to merge gVCF records from each individual *using* the multi-sample joint aggregation step that combines all records, generates correct genotype likelihoods, re-genotypes the newly merged records and reannotates each of the called variants²⁵. Raw variant calls were then filtered to retain only monomorphic and biallelic Single Nucleotide Polymorphisms (SNPs) for downstream analyses.

Genotype filtering: We excluded genotype calls from repetitive regions detected using Repeat Masker v4.1.0 and genomic regions with a mappability score <1. Mappability was calculated using GENMAP⁷² (<https://github.com/cpockrandt/genmap>) using a k-mer length of 100 and a maximum of two mismatches. Second, we excluded any genotype call with a genotype quality (GQ) score <20, on the basis that genotype accuracy rapidly declines below this threshold (see GATK manual). Further, we removed sites where either parental genotype was missing as these are not informative. Following this, we extracted a subset of sites where parents were homozygous for different alleles and all nine offspring were heterozygous (genotype calls were considered homozygous in parents if the minor allele balance was <0.1 and heterozygous in offspring if the minor allele balance was ≥ 0.25). From these high-confidence heterozygous sites we extracted the following quality annotations from the VCF INFO field: base quality rank sum; read position rank sum; mapping quality; and quality by depth. In addition, from the VCF FORMAT field we extracted depth of coverage annotations for individual genotypes. We then examined their distributions in the high confidence heterozygous sites and used the 5th and 95th percentiles calculated for each quality annotation as standard cut-offs to filter biallelic and monomorphic sites in our entire dataset. Note: for mapping quality and quality by depth we only applied the lower cut-off (5th percentile) to prevent penalisation of high-quality sites. Sites

Formatted: Font: (Default) Times New Roman

that passed our in-house filtering pipeline were considered high confidence “callable” sites.

De novo mutation calling. From the filtered dataset generated in the previous step, we identified candidate *de novo* mutations as sites where both parents were homozygous for the same allele and at least one offspring carried a variant allele in the heterozygous state. We conducted secondary genotype calling at these positions, using the *mpileup* and *call* functions of bcftools⁷³ v1.14 (<https://samtools.github.io/bcftools/>), and considered sites as true candidates when GATK and bcftools genotypes matched, and when the putative mutation was supported by at least 25% reads i.e. had a minor allele balance ≥ 0.25 .

Estimation of the false negative rate. We simulated mutations by introducing variants directly into sample BAM files using the Single Nucleotide Variant (SNV) simulation tool SomatoSim³⁰ (<https://github.com/BieseckerLab/SomatoSim>). The advantages of this approach compared to generating synthetic reads from a reference file is that this approach allows for error profiles to be preserved and does not limit variant allele frequencies (VAFs), variant locations, or the number of variants that can be simulated. For a single offspring (ind1722) we simulated mutations at 947 invariant sites within high confidence callable regions. Each mutated site had its frequency of mutated reads determined by sampling from the observed frequency distribution of callable heterozygous sites in the original dataset. We then repeated previously described genotype calling, genotype filtering, and *de novo* mutation detection pipelines, and compared the SNP calls with expected genotypes based on the mutated sites and calculated the false negative rate.

Experimental validation and parental origin of *de novo* mutations

To confirm the authenticity of candidate *de novo* mutations, we performed Sanger sequencing of the genomic regions around each candidate in both parents and all nine offspring. To confirm the sequence of the parents at candidate mutation sites genomic DNA was extracted from blood samples using a PureLink™ Genomic DNA Mini Kit (ThermoFisher) and used as template for PCR amplification. PCR was performed using Phusion® High Fidelity DNA Polymerase (NEB) or PrimeSTAR GXL DNA Polymerase (Takara). Primer pairs are summarized in Supplementary Table 3. Sanger sequencing was performed on amplified fragments with the respective forward and reverse PCR primers or fragments were cloned into pGEM®Teasy vector (Promega) and sequenced from M13 forward and M13 reverse sites.

Commented [A5]: The way this is written, it seems like the authors look at mutations like the following:

Mom CAAT/CAAT
Dad CAAT/CAAT

Offspring: CAAT/CAAA

Why just these mutations? Why not looking at others, e.g.:

Mom CAAT/CAAT
Dad CAAA/CAAA

Offspring: CAAT/CAAG or CAAA/CAAG

It seems like you will miss mutations and underestimate the mutation rate. This is obviously at the heart of the mutation rate calculation and a better justification is needed regarding why only certain mutations were considered.

Formatted: Font: (Default) Times New Roman

For mutations detected in offspring individuals the same DNA samples used for whole genome sequencing were used. PCR and Sanger sequence-based screen PCR amplification of the region of interest was performed in a total volume of 10 μ l making use of the Phusion Flash Mastermix (Thermo Scientific) with 2 μ l input of genomic DNA, 0.5 % DMSO and 0.5 μ M forward and reverse primer. All details on primer sequences (target-seq-primer) and on PCR conditions are listed in Supplementary Table 3. Sanger sequencing was performed either with the respective forward and reverse PCR primer and if required with internally located dedicated sequencing primers to cover the region of interest.

Parental origin of *de novo* mutations

We attempted to infer the parental offspring of verified *de novo* mutations based on the occurrence of flanking SNP alleles that segregated between the two parents (i.e., positions where parents were homozygous for different alleles) and occurred within the same Illumina read or mate-pair read. Due to the limited presence of segregating sites, this was only possible for a single *de novo* mutation.

Estimation of nucleotide diversity and effective population size

For parental samples we estimated nucleotide diversity (π) in non-overlapping 100 Kb windows using pixy⁷⁴. Given the relationship between nucleotide diversity (π), mutation rate (μ) and effective population size (N_e) for diploid organisms is $\pi = 4N_e\mu$, we extrapolated the effective population size using the formula: $N_e = \pi/4\mu$.

ACKNOWLEDGEMENTS

We gratefully acknowledge the contribution of the Long Read Team of the Dresden Concept Genome Center. This study was supported by the DFG Research Infrastructure NGS-CC as part of the Next Generation Sequencing Competence Network (project 423957469) and grants from the Deutsche Forschungsgemeinschaft (SCHA 408/15-1) as part of the DFG Sequencing call to MS, Vetenskapsrådet (2017-02907) and Knut and Alice Wallenberg Foundation (KAW 2016.0361) to LA, Australian Research Council Discovery Grant DP220102970 to PDC and FT, Florida Museum of Natural History to GN, and an NHMRC Fellowship GNT1136567 to PDC. This research was supported in part by the Intramural Research Program of the National Human Genome Research Institute (ZIAHG200386-06). The authors would like to

acknowledge the use of computing resources at Uppsala Multidisciplinary Center for Advanced Computational Science (UPPMAX) in carrying out this work.

AUTHOR CONTRIBUTIONS

LA, MS conceived and supervised the study; FT, PC, RD provided the biological material; AP, OF, JM, JB, BH, GN, AD, EJ, SB generated the reference genome assembly, AD, MM prepared the karyotype; FT, PC provided transcriptome sequence; DK performed the annotation; SW, GM sequenced the offspring; ATSP, MP analysed the pedigree data and identified candidate mutations; SW, FT, KK, and ATSP validated the mutations; LA, PC, SB, MS interpreted the data; ATSP drafted the manuscript; SB, PC, LA revised the manuscript with input from other authors; all authors approved the manuscript prior to submission.

COMPETING INTERESTS

The authors declare no competing interest.

DATA AVAILABILITY

Data used for genome assembly construction are available at: https://genomeark.github.io/genomeark-all/Hemiscyllium_ocellatum/. Additional raw Illumina sequencing reads used for detection of candidate *de novo* mutations have been deposited at NCBI under the BioProject PRJNA900175. Genome assemblies are available from NCBI under GenBank accession numbers GCA_020745735.1 (paternal haplotype) and GCA_020745765.1 (maternal haplotype). Both haplotypes are also available through the UCSC genome browser gateway (<https://genome.ucsc.edu>).

CODE AVAILABILITY

Code used for the detection of candidate *de novo* mutations and estimation of mutation rate are available at: https://github.com/LeifAnderssonLab/Epaulette_shark_mutation_rate.

REFERENCES

1. Compagno, L. J. V. Alternative life-history styles of cartilaginous fishes in time and space. *Environ. Biol. Fishes* **28**, 33–75 (1990).

Commented [A6]: Make sure these all have consistent formatting

2. Kriwet, J., Witzmann, F., Klug, S. & Heidtke, U. H. J. First direct evidence of a vertebrate three-level trophic chain in the fossil record. *Proc. Biol. Sci.* **275**, 181–186 (2008).
3. Ferretti, F., Worm, B., Britten, G. L., Heithaus, M. R. & Lotze, H. K. Patterns and ecosystem consequences of shark declines in the ocean. *Ecol. Lett.* **13**, 1055–1071 (2010).
4. Heithaus, M. R., Wirsing, A. J. & Dill, L. M. The ecological importance of intact top-predator populations: a synthesis of 15 years of research in a seagrass ecosystem. *Mar. Freshwater Res.* **63**, 1039–1050 (2012).
5. Stevens, J. D., Bonfil, R., Dulvy, N. K. & Walker, P. A. The effects of fishing on sharks, rays, and chimaeras (chondrichthyans), and the implications for marine ecosystems. *ICES J. Mar. Sci.* **57**, 476–494 (2000).
6. Oliver, S., Braccini, M., Newman, S. J. & Harvey, E. S. Global patterns in the bycatch of sharks and rays. *Mar. Policy* **54**, 86–97 (2015).
7. Clarke, S., Milner-Gulland, E. J. & Bjørndal, T. Social, Economic, and Regulatory Drivers of the Shark Fin Trade. *Mar. Resour. Econ.* **22**, 305–327 (2007).
8. William Lane, I. & Comac, L. *Sharks Still Don't Get Cancer*. (Avery Publishing Group, 1996).
9. William Lane, I. *Sharks Don't Get Cancer*. (Avery Pub., 1992).
10. Patra, D. & Sandell, L. J. Antiangiogenic and anticancer molecules in cartilage. *Expert Rev. Mol. Med.* **14**, e10 (2012).
11. Ostrander, G. K., Cheng, K. C., Wolf, J. C. & Wolfe, M. J. Shark cartilage, cancer and the growing threat of pseudoscience. *Cancer Res.* **64**, 8485–8491 (2004).
12. Cortés, E. Life History Patterns and Correlations in Sharks. *Rev. Fish. Sci.* **8**, 299–344 (2000).
13. Musick, J. A. Life in the Slow Lane: Ecology and conservation of long-lived marine animals. *American Fisheries Society, Maryland* (1999).
14. Cortés, E. Incorporating Uncertainty into Demographic Modeling: Application to Shark Populations and Their Conservation. *Conserv. Biol.* **16**, 1048–1062 (2002).
15. García, V. B., Lucifora, L. O. & Myers, R. A. The importance of habitat and life history to extinction risk in sharks, skates, rays and chimaeras. *Proc. Biol. Sci.* **275**, 83–89 (2008).
16. Dulvy, N. K. & Forrest, R. E. Life histories, population dynamics, and extinction risks in

chondrichthyans. in *Sharks and their relatives II* 655–696 (CRC Press, 2010).

17. Bergeron, L. A. *et al.* Evolution of the germline mutation rate across vertebrates. *Nature* (2023) doi:10.1038/s41586-023-05752-y.
18. Hara, Y. *et al.* Shark genomes provide insights into elasmobranch evolution and the origin of vertebrates. *Nat Ecol Evol* **2**, 1761–1771 (2018).
19. Martin, A. P., Naylor, G. J. P. & Palumbi, S. R. Rates of mitochondrial DNA evolution in sharks are slow compared with mammals. *Nature* **357**, 153–155 (1992).
20. Dudgeon, C. L. *et al.* Walking, swimming or hitching a ride? Phylogenetics and biogeography of the walking shark genus *Hemiscyllium*. *Mar. Freshwater Res.* **71**, 1107–1117 (2020).
21. Heupel, M. R., Whittier, J. M. & Bennett, M. B. Plasma steroid hormone profiles and reproductive biology of the epaulette shark, *Hemiscyllium ocellatum*. *J. Exp. Zool.* **284**, 586–594 (1999).
22. Rhie, A. *et al.* Towards complete and error-free genome assemblies of all vertebrate species. *Nature* **592**, 737–746 (2021).
23. Koren, S. *et al.* De novo assembly of haplotype-resolved genomes with trio binning. *Nat. Biotechnol.* (2018) doi:10.1038/nbt.4277.
24. Koren, S. *et al.* Canu: scalable and accurate long-read assembly via adaptive k-mer weighting and repeat separation. *Genome Res.* **27**, 722–736 (2017).
25. McKenna, A. *et al.* The Genome Analysis Toolkit: A MapReduce framework for analyzing next-generation DNA sequencing data. *Genome Res.* **20**, 1297–1303 (2010).
26. Yoder, A. D. & Tiley, G. P. The challenge and promise of estimating the de novo mutation rate from whole-genome comparisons among closely related individuals. *Mol. Ecol.* **30**, 6087–6100 (2021).
27. Feng, C. *et al.* Moderate nucleotide diversity in the Atlantic herring is associated with a low mutation rate. *Elife* **6**, e23907 (2017).
28. Koch, E. M. *et al.* De Novo Mutation Rate Estimation in Wolves of Known Pedigree. *Mol. Biol. Evol.* **36**, 2536–2547 (2019).
29. Vogel, F. & Kopun, M. Higher frequencies of transitions among point mutations. *J. Mol. Evol.* **9**,

159–180 (1977).

30. Hawari, M. A., Hong, C. S. & Biesecker, L. G. SomatoSim: precision simulation of somatic single nucleotide variants. *BMC Bioinformatics* **22**, 109 (2021).
31. Pearce, J., Fraser, M. W., Sequeira, A. M. M. & Kaur, P. State of shark and ray genomics in an era of extinction. *Front. Mar. Sci.* **8**, (2021).
32. Martin, A. P. & Palumbi, S. R. Body size, metabolic rate, generation time, and the molecular clock. *Proc. Natl. Acad. Sci. U. S. A.* **90**, 4087–4091 (1993).
33. Whitney, N. M., Lear, K. O., Gaskins, L. C. & Gleiss, A. C. The effects of temperature and swimming speed on the metabolic rate of the nurse shark (*Ginglymostoma cirratum*, Bonaterre). *J. Exp. Mar. Bio. Ecol.* **477**, 40–46 (2016).
34. White, C. R. & Seymour, R. S. Allometric scaling of mammalian metabolism. *J. Exp. Biol.* **208**, 1611–1619 (2005).
35. MacNeil, M. A. *et al.* Biology of the Greenland shark *Somniosus microcephalus*. *J. Fish Biol.* **80**, 991–1018 (2012).
36. Ste-Marie, E., Watanabe, Y. Y., Semmens, J. M., Marcoux, M. & Hussey, N. E. A first look at the metabolic rate of Greenland sharks (*Somniosus microcephalus*) in the Canadian Arctic. *Sci. Rep.* **10**, 19297 (2020).
37. Langer, R., Brem, H., Falterman, K., Klein, M. & Folkman, J. Isolations of a cartilage factor that inhibits tumor neovascularization. *Science* **193**, 70–72 (1976).
38. Lee, A. & Langer, R. Shark cartilage contains inhibitors of tumor angiogenesis. *Science* **221**, 1185–1187 (1983).
39. Camhi, M. D., Valenti, S. V., Fordham, S. V., Fowler, S. L. & Gibson, C. The conservation status of pelagic sharks and rays: report of the IUCN shark specialist group pelagic shark red list workshop. *IUCN Species Survival Commission Shark Specialist Group. Newbury, UK. x+ 78p* (2009).
40. Horsman, M. R., Alsner, J. & Overgaard, J. The effect of shark cartilage extracts on the growth and metastatic spread of the SCCVII carcinoma. *Acta Oncol.* **37**, 441–445 (1998).
41. Miller, D. R., Anderson, G. T., Stark, J. J., Granick, J. L. & Richardson, D. Phase I/II trial of the

Formatted: Font: Italic

Formatted: Font: Italic

Formatted: Font: Italic

- safety and efficacy of shark cartilage in the treatment of advanced cancer. *J. Clin. Oncol.* **16**, 3649–3655 (1998).
42. Lu, C. *et al.* Chemoradiotherapy With or Without AE-941 in Stage III Non–Small Cell Lung Cancer: A Randomized Phase III Trial. *J. Natl. Cancer Inst.* **102**, 859–865 (2010).
 43. Loprinzi, C. L. *et al.* Evaluation of shark cartilage in patients with advanced cancer: a North Central Cancer Treatment Group trial. *Cancer* **104**, 176–182 (2005).
 44. Cannataro, V. L., Mandell, J. D. & Townsend, J. P. Attribution of Cancer Origins to Endogenous, Exogenous, and Preventable Mutational Processes. *Mol. Biol. Evol.* **39**, (2022).
 45. Qing, T. *et al.* Germline variant burden in cancer genes correlates with age at diagnosis and somatic mutation burden. *Nat. Commun.* **11**, 2438 (2020).
 46. Milholland, B. *et al.* Differences between germline and somatic mutation rates in humans and mice. *Nat. Commun.* **8**, 15183 (2017).
 47. Heupel, M. R. & Bennett, M. B. Estimating Abundance of Reef-Dwelling Sharks: A Case Study of the Epaulette Shark, *Hemiscyllium ocellatum* (Elasmobranchii: Hemiscyllidae). *J. pasc* **61**, 383–394 (2007).
 48. Springer, V. G., Last, P. R. & Stevens, J. D. Sharks and rays of Australia. *Copeia* **1994**, 1055 (1994).
 49. Allen, G. R., Erdmann, M. V., White, W. T., Dudgeon, C. L. & Others. Review of the bamboo shark genus *Hemiscyllium* (Orectolobiformes: Hemiscyllidae). *J. Ocean Sci. Found.* **23**, 51–97 (2016).
 50. Lynch, M. Evolution of the mutation rate. *Trends Genet.* **26**, 345–352 (2010).
 51. Ballard, W. W., Mellinger, J. & Lechenault, H. A series of normal stages for development of *Scyliorhinus canicula*, the lesser spotted dogfish (Chondrichthyes: Scyliorhinidae). *J. Exp. Zool.* **267**, 318–336 (1993).
 52. Onimaru, K., Motone, F., Kiyatake, I., Nishida, K. & Kuraku, S. A staging table for the embryonic development of the brownbanded bamboo shark (*Chiloscyllium punctatum*). *Dev. Dyn.* **247**, 712–723 (2018).
 53. Guan, D. *et al.* Identifying and removing haplotypic duplication in primary genome assemblies.

Formatted: Font: Italic

Formatted: Font: Italic

Formatted: Font: Italic

Formatted: Font: Italic

Bioinformatics **36**, 2896–2898 (2020).

54. Ghurye, J. *et al.* Integrating Hi-C links with assembly graphs for chromosome-scale assembly. *PLoS Comput. Biol.* **15**, e1007273 (2019).
55. Li, H. Minimap2: pairwise alignment for nucleotide sequences. *Bioinformatics* **34**, 3094–3100 (2018).
56. Garrison, E. & Marth, G. Haplotype-based variant detection from short-read sequencing. *arXiv [q-bio.GN]* (2012).
57. Formenti, G. *et al.* Merfin: improved variant filtering, assembly evaluation and polishing via k-mer validation. *Nat. Methods* **19**, 696–704 (2022).
58. Rhie, A., Walenz, B. P., Koren, S. & Phillippy, A. M. Merqury: reference-free quality, completeness, and phasing assessment for genome assemblies. *Genome Biol.* **21**, 245 (2020).
59. Howe, K. *et al.* Significantly improving the quality of genome assemblies through curation. *Gigascience* **10**, (2021).
60. Formenti, G. *et al.* Complete vertebrate mitogenomes reveal widespread repeats and gene duplications. *Genome Biol.* **22**, 120 (2021).
61. Du, K. *et al.* Genome biology of the darkedged splitfin, *Girardinichthys multiradiatus*, and the evolution of sex chromosomes and placentation. *Genome Res.* **32**, 583–594 (2022).
62. Slater, G. S. C. & Birney, E. Automated generation of heuristics for biological sequence comparison. *BMC Bioinformatics* **6**, 31 (2005).
63. Birney, E., Clamp, M. & Durbin, R. GeneWise and Genomewise. *Genome Res.* **14**, 988–995 (2004).
64. Kim, D., Langmead, B. & Salzberg, S. L. HISAT: a fast spliced aligner with low memory requirements. *Nat. Methods* **12**, 357–360 (2015).
65. Pertea, M. *et al.* StringTie enables improved reconstruction of a transcriptome from RNA-seq reads. *Nat. Biotechnol.* **33**, 290–295 (2015).
66. Grabherr, M. G. *et al.* Full-length transcriptome assembly from RNA-Seq data without a reference genome. *Nat. Biotechnol.* **29**, 644–652 (2011).
67. Kapustin, Y., Souvorov, A., Tatusova, T. & Lipman, D. Splign: algorithms for computing spliced

Formatted: Font: Italic

- alignments with identification of paralogs. *Biol. Direct* **3**, 20 (2008).
68. Stanke, M. *et al.* AUGUSTUS: ab initio prediction of alternative transcripts. *Nucleic Acids Res.* **34**, W435–9 (2006).
 69. Simão, F. A., Waterhouse, R. M., Ioannidis, P., Kriventseva, E. V. & Zdobnov, E. M. BUSCO: assessing genome assembly and annotation completeness with single-copy orthologs. *Bioinformatics* **31**, 3210–3212 (2015).
 70. Uno, Y. *et al.* Cell culture-based karyotyping of orectolobiform sharks for chromosome-scale genome analysis. *Commun Biol* **3**, 652 (2020).
 71. Vasimuddin, M., Misra, S., Li, H. & Aluru, S. Efficient Architecture-Aware Acceleration of BWA-MEM for Multicore Systems. in *2019 IEEE International Parallel and Distributed Processing Symposium (IPDPS)* 314–324 (2019).
 72. Pockrandt, C., Alzamel, M., Iliopoulos, C. S. & Reinert, K. GenMap: ultra-fast computation of genome mappability. *Bioinformatics* **36**, 3687–3692 (2020).
 73. Danecek, P. *et al.* Twelve years of SAMtools and BCFtools. *Gigascience* **10**, giab008 (2021).
 74. Korunes, K. L. & Samuk, K. pixy: Unbiased estimation of nucleotide diversity and divergence in the presence of missing data. *Mol. Ecol. Resour.* **21**, 1359–1368 (2021).
 75. Challis, R., Richards, E., Rajan, J., Cochrane, G. & Blaxter, M. BlobToolKit – Interactive Quality Assessment of Genome Assemblies. *G3 Genes|Genomes|Genetics* **10**, 1361–1374 (2020).
 76. Malinsky, M. *et al.* Whole-genome sequences of Malawi cichlids reveal multiple radiations interconnected by gene flow. *Nature Ecology & Evolution* **2**, 1940–1955 (2018).
 77. Smeds, L., Qvarnström, A. & Ellegren, H. Direct estimate of the rate of germline mutation in a bird. *Genome Res.* **26**, 1211–1218 (2016).
 78. Harland, C. *et al.* Frequency of mosaicism points towards mutation-prone early cleavage cell divisions in cattle. *bioRxiv* 079863 (2017).
 79. Zhang, M., Yang, Q., Ai, H. & Huang, L. Revisiting the evolutionary history of pigs via De Novo mutation rate estimation in a three-generation pedigree. *Genomics Proteomics Bioinformatics* (2022) doi:10.1016/j.gpb.2022.02.001.

80. Wang, R. J. *et al.* De novo mutations in domestic cat are consistent with an effect of reproductive longevity on both the rate and spectrum of mutations. *bioRxiv* 2021.04.06.438608 (2021).
81. Martin, H. C. *et al.* Insights into Platypus Population Structure and History from Whole-Genome Sequencing. *Mol. Biol. Evol.* **35**, 1238–1252 (2018).
82. Thomas, G. W. C. *et al.* Reproductive Longevity Predicts Mutation Rates in Primates. *Curr. Biol.* **28**, 3193–3197.e5 (2018).
83. Yang, C. *et al.* Evolutionary and biomedical insights from a marmoset diploid genome assembly. *Nature* **594**, 227–233 (2021).
84. Pfeifer, S. P. Direct estimate of the spontaneous germ line mutation rate in African green monkeys. *Evolution* **71**, 2858–2870 (2017).
85. Besenbacher, S., Hvilsom, C., Marques-Bonet, T., Mailund, T. & Schierup, M. H. Direct estimation of mutations in great apes reconciles phylogenetic dating. *Nature Ecology & Evolution* **3**, 286–292 (2019).
86. Conrad, D. F. *et al.* Variation in genome-wide mutation rates within and between human families. *Nat. Genet.* **43**, 712–714 (2011).
87. Kong, A. *et al.* Rate of de novo mutations, father's age, and disease risk. *Nature* **488**, 471–475 (2012).
88. Francioli, L. C. *et al.* Genome-wide patterns and properties of de novo mutations in humans. *Nat. Genet.* **47**, 822–826 (2015).
89. Rahbari, R. *et al.* Timing, rates and spectra of human germline mutation. *Nat. Genet.* **48**, 126–133 (2016).
90. Wong, W. S. W. *et al.* New observations on maternal age effect on germline de novo mutations. *Nat. Commun.* **7**, 10486 (2016).
91. Jónsson, H. *et al.* Parental influence on human germline de novo mutations in 1,548 trios from Iceland. *Nature* **549**, 519–522 (2017).
92. Maretty, L. *et al.* Sequencing and de novo assembly of 150 genomes from Denmark as a population reference. *Nature* **548**, 87–91 (2017).
93. Turner, T. N. *et al.* Genomic Patterns of De Novo Mutation in Simplex Autism. *Cell* **171**, 710–

722.e12 (2017).

94. Sasani, T. A. *et al.* Large, three-generation human families reveal post-zygotic mosaicism and variability in germline mutation accumulation. *Elife* **8**, e46922 (2019).
95. Kessler, M. D. *et al.* De novo mutations across 1,465 diverse genomes reveal mutational insights and reductions in the Amish founder population. *Proc. Natl. Acad. Sci. U. S. A.* **117**, 2560–2569 (2020).
96. Wang, R. J. *et al.* Paternal age in rhesus macaques is positively associated with germline mutation accumulation but not with measures of offspring sociability. *Genome Res.* **30**, 826–834 (2020).
97. Bergeron, L. A. *et al.* The germline mutational process in rhesus macaque and its implications for phylogenetic dating. *Gigascience* **10**, giab029 (2021).
98. Campbell, C. R. *et al.* Pedigree-based and phylogenetic methods support surprising patterns of mutation rate and spectrum in the gray mouse lemur. *Heredity* **127**, 233–244 (2021).
99. Venn, O. *et al.* Nonhuman genetics. Strong male bias drives germline mutation in chimpanzees. *Science* **344**, 1272–1275 (2014).
100. Tatsumoto, S. *et al.* Direct estimation of de novo mutation rates in a chimpanzee parent-offspring trio by ultra-deep whole genome sequencing. *Sci. Rep.* **7**, 13561 (2017).
101. Wu, F. L. *et al.* A comparison of humans and baboons suggests germline mutation rates do not track cell divisions. *PLoS Biol.* **18**, e3000838 (2020).
102. Lindsay, S. J., Rahbari, R., Kaplanis, J., Keane, T. & Hurles, M. E. Similarities and differences in patterns of germline mutation between mice and humans. *Nat. Commun.* **10**, 4053 (2019).

FIGURES AND TABLES

Fig. 1. Distribution and brood colony of Epaulette sharks (*Hemiscyllium ocellatum*). (a) Map showing epaulette shark geographic distribution according to refs²⁰ and ⁴⁹. (b) Isolated male and female adults used in this study. (c) Stage 37 pre-hatchling removed from egg case. Small remaining yolk ball is present on the right.

Fig. 2. Genome assembly metrics. BlobToolKit⁷⁵ snailplot showing N50 metrics and BUSCO gene completeness of the paternal genome assembly. The main plot is divided into 1,000 size-ordered bins around the circumference with each bin representing 0.1% of the assembly. The distribution of scaffold lengths is shown in dark grey with the plot radius scaled to the longest scaffold present in the assembly (shown in red). Orange and pale-orange arcs show the N50 and N90 scaffold lengths, respectively. The pale grey spiral shows the cumulative scaffold count on a log scale with white scale lines showing successive orders of magnitude. The blue and pale-blue area around the outside of the plot shows the distribution of GC, AT and N percentages in the same bins as the inner plot. A summary of complete, fragmented, duplicated and missing BUSCO genes in the vertebrata_odb9 set is shown in the top right. See Supplementary Fig. 1 for snailplot of the maternal genome assembly.

Fig. 3. Genotype filtering and *de novo* mutation-calling pipeline used in this study.

Fig. 4. Identification of candidate *de novo* mutations (DNMs). (a) Sample genotypes at positions containing candidate *de novo* mutations. High confidence genotypes are indicated by black text, and low confidence genotypes indicated by grey text. Genotypes were considered “high confidence” when GATK and bcftools derived genotypes matched. Parental genotypes and offspring candidate *de novo* mutations are coloured according to their validation status. (b) Chromatograms showing parental and focal offspring Sanger sequences for confirmed *de novo* mutations. Additional chromatograms are presented in Extended Data Figures 2-5.

a

Candidate DNM		Parental genotype		Offspring genotype								
Scaffold	Position	sHemOce2	sHemOce3	ind1722	ind1835	ind1895	ind1923	ind1925	ind1983	ind2023	ind2024	ind2046
scaffold_4_mat	13524629	AA	AA	AG	AG	AG	AG	AG	AG	AG	AG	AG
scaffold_4_mat	48132425	GG	GG	GG	GG	GG	GG	GG	GG	GA	GG	GG
scaffold_8_mat	66018494	TT	TT	TT	TT	TT	TT	TT	TC	TT	TT	TT
scaffold_14_mat	4777531	AA	AA	AG	AG	AG	AG	AG	AG	AG	AG	AG
scaffold_19_mat	62722441	TT	TT	TT	TT	TT	TT	TC	TT	TT	TT	TT
scaffold_24_mat	38442575	GG	GG	AG	AG	AG	GG	AG	GG	AG	AG	GG
scaffold_28_mat	17411576	AA	AA	AA	AA	AA	AA	AA	AA	AT	AA	AA
scaffold_36_mat	8326154	GG	GG	GG	GG	AG	GG	GG	GG	AG	GG	GG
scaffold_51_mat	8719242	CC	CC	CC	CT	CC	CC	CC	CC	CC	CC	CC
scaffold_158_mat	387249	GG	GG	GG	GG	GG	GG	GG	GA	GG	GG	GG

GG Genotype passed in-house filtering GG Genotype failed in-house filtering Genotype confirmed Genotype dismissed Sanger sequencing failed filtering

Fig. 5. The distribution of nucleotide diversity (π) values estimated in nonoverlapping 100kb windows for parental samples (sHemOce2 and sHemOce3). The dashed line indicates mean π calculated across both samples.

Fig. 6. Directly estimated vertebrate *de novo* mutation rates. Where multiple estimates are available for a given species, see Supplementary Table 1, the mean is presented. Note: only mutation rate estimates that include validation of mutations via Sanger sequencing are reported.

Extended Data Fig. 1. Karyotype of the epaulette shark. The Epaulette shark karyotype has 52 pairs of autosomes plus X and Y sex chromosomes (53, XY). This includes six metacentric chromosome (pairs: 2, 10, 17, 19, 29, X); 13 submetacentric chromosomes (pairs: 3, 4, 5, 6, 8, 11, 13, 30, 39, 40, 43, 44, 45); 24 acrocentric chromosomes (pairs: 1, 7, 9, 12, 15, 16, 18, 20, 21, 22, 23, 24, 26, 27, 33, 34, 35, 38, 46, 47, 48, 49, 51, Y); and 11 telocentric chromosomes (pairs: 14, 25, 28, 31, 32, 36, 37, 41, 42, 50, 52).

Extended Data Fig. 2. Chromatograms showing parental and focal offspring Sanger sequences around: **(a)** the candidate *de novo* mutation identified in ind2023 at scaffold_4_mat: 48132425 bp; and **(b)** the candidate *de novo* mutation identified in ind1983 at scaffold_8_mat: 66018494 bp.

Extended Data Fig. 3. Chromatograms showing parental and focal offspring Sanger sequences around: **(a)** the candidate *de novo* mutation identified in ind1925, ind2023 and ind2024 at scaffold_14_mat: 4777531 bp; and **(b)** the candidate *de novo* mutation identified in ind1925 at scaffold_19_mat: 62722441 bp.

Extended Data Fig. 4. Chromatograms showing parental and focal offspring Sanger sequences around: **(a)** the candidate *de novo* mutation identified in ind2023 at scaffold_24_mat: 38442575 bp; and **(b)** the candidate *de novo* mutation identified in ind1923 at scaffold_36_mat: 8326154 bp.

Extended Data Fig. 5. Chromatograms showing parental and focal offspring Sanger sequences around: **(a)** the candidate *de novo* mutation identified in ind1835 at scaffold_51_mat: 8719424 bp; and **(b)** the candidate *de novo* mutation identified in ind1983 at scaffold_158_mat: 387249 bp.

Extended Data Table 1. Statistics of the genome assembly

Total sequence length	3,983,483,121
Total ungapped length	3,905,918,519
Gaps between scaffolds	0
Number of scaffolds	1,963
Scaffold N50	83,580,160
Scaffold L50	17
Number of contigs	5,667
Contig N50	8,845,575
Contig L50	92
Total number of chromosomes and plasmids	55
Number of component sequences (WGS or clone)	1,963

Extended Data Table 2. Metrics of the genome annotation. C=complete, S=single, D=duplicate, F=fragmented, M=missing.

Total_# of genes	18,225
Single_exon genes	1,275 (7.0%)
Multi_exon genes	16,950 (93.0%)
Genes mapping_to_Pfam	16,495 (90.5%)
Genes mapping_to_RefSeq/SwissProt	17,580 (96.5%)
Genes with_RNA_support	15,411 (84.6%)
Genes without_start_codon	1,090 (6.0%)
Pseudogenes_with_RNA	732 (4.0%)
Pseudogenes_without_RNA	520 (2.9%)
BUSCO analysis using the vertebrata_odb9 gene set (n=2,586)	
Before annotation	C:89.3%[S:86.8%,D:2.5%],F:4.4%,M:6.3%
After annotation	C:96.0%[S:92.0%,D:4.0%],F:2.2%,M:1.8%

SUPPLEMENTARY MATERIALS

**Pedigree-based estimate of de novo mutation rate in the epaulette shark
defines the lowest known vertebrate mutation rate**

Supplementary Fig. 1. Genome assembly metrics. BlobToolKit⁷⁵ snailplot showing N50 metrics and BUSCO gene completeness of the maternal genome assembly. The main plot is divided into 1,000 size-ordered bins around the circumference with each bin representing 0.1% of the assembly. The distribution of scaffold lengths is shown in dark grey with the plot radius scaled to the longest scaffold present in the assembly (shown in red). Orange and pale-orange arcs show the N50 and N90 scaffold lengths, respectively. The pale grey spiral shows the cumulative scaffold count on a log scale with white scale lines showing successive orders of magnitude. The blue and pale-blue area around the outside of the plot shows the distribution of GC, AT and N percentages in the same bins as the inner plot. A summary of complete, fragmented, duplicated and missing BUSCO genes in the vertebrata_odb9 set is shown in the top right.

Supplementary Fig. 2. The *de novo* mutation identified on scaffold_8_mat (position: 66018494bp) can be determined to be of paternal origin based on the presence of a downstream SNP that is segregating in parental samples. In the focal offspring (ind1983) reads spanning both sites that contain a mutated allele (C) always contain the paternal allele (C) at the segregating site.

Supplementary Table 1.

Directly estimated vertebrate *de novo* mutation rates with PCR confirmation used to generate Fig. 6.

Class	Order	Species	Common Name	Ref	No. trios	Rate ^a
Actinopterygii	Cichliformes	Astatotilapia calliptera	Eastern river bream	⁷⁶	9	35
		Aulonocara stuartgranti	Flavescent peacock	⁷⁶	9	35
		Lethrinops lethrinus	Scarlet Fin Lethrinus	⁷⁶	9	35
	Clupeiformes	Clupea harengus	Atlantic herring	²⁷	12	20
Aves	Passeriformes	Ficedula albicollis	Collared flycatcher	⁷⁷	7	46
Chondrichthyes	Orectolobiformes	Hemiscyllium ocellatum	Epaulette shark	This work	9	6.8
Mammalia	Artiodactyla	Bos taurus	Cattle	⁷⁸	5	117
	Artiodactyla	Sus scrofa	Wild boar	⁷⁹	5	36
	Carnivora	Canis lupus	Wolf	²⁸	4	45
		Felis catus	Domestic cat	⁸⁰	11	86
	Monotremata	Ornithorhynchus anatinus	Platypus	⁸¹	2	70
	Primates	Aotus nancymae	Owl monkey	⁸²	14	81
		Callithrix jacchus	Marmoset	⁸³	1	43
		Chlorocebus sabaeus	Green monkey	⁸⁴	3	94

		Gorilla gorilla	Gorilla	85	2	113
		Homo sapiens	Human	86	1	117
				86	1	97
				87	78	120
				88	269	120
				89	13	128
				90	719	105
				91	1550	129
				92	150	128
				93	516	130
				94	593	110
		95	1449	122		
		Macaca mulatta	Rhesus macaque	96	14	58
				97	19	77
		Microcebus murinus	Grey mouse lemur	98	2	152
		Pan troglodytes	Chimpanzee	99	6	120
				100	1	148

				85	7	126
		Papio anubis	Baboon	101	12	57
		Pongo abelii	Orangutan	85	1	166
	Rodentia	Mus musculus	House mouse	46	8	57
				102	15	39

^a Mutation rate per site per generation: $\mu \times 10^{-10}$

Supplementary Table 2.

Summary of the pedigree and the sequencing depth used for estimation of the *de novo* mutation rate.

Sample ID	Pedigree	Sequencing depth (x)	Callable 'informative' sites per trio	Accession no.
sHemOce2	Mother	113.0	-	
sHemOce3	Father	135.6	-	
ind1722	Offspring	82.0	415,534,410	pending
ind1835	Offspring	74.7	431,947,407	pending
ind1895	Offspring	49.9	426,853,267	pending
ind1923	Offspring	95.9	333,716,129	pending
ind1925	Offspring	61.9	457,044,582	pending
ind1983	Offspring	60.4	426,022,109	pending
ind2023	Offspring	70.1	382,847,039	pending
ind2024	Offspring	75.9	386,355,463	pending
ind2046	Offspring	60.8	431,490,538	pending

Supplementary Table 3.

Primer pairs used to amplify genomic regions containing candidate *de novo* mutations from the parents and offspring

Supplementary table 3a: Primers for offspring

Target candidate de novo mutation	Primer set	Fragment size	Primer name	Tm	Sequence 5' - 3'
Scaffold_4_mat: 48132425	PCR primer set	615 bp -	1861_HemOce-sc4-3-F	60.07	AAT TCG AAG GAC CGC AGA TGT
			1862_HemOce-sc4-3-R	58.86	AAG GCA GAA CAG AAG TCC CA
	Sequencing primer		1861_HemOce-sc4-3-F	-	AAT TCG AAG GAC CGC AGA TGT
Scaffold_4_mat: 13524629	PCR primer set 1	638 bp	1920_HemOce-sc4-5-F	59.67	CTG CTG CTG CTG AGA TTT ACG
			1921_HemOce-sc4-5-R	59.79	TCT TAT CTG TCA GCC AAG GGC
	Sequencing primer 1		1920_HemOce-sc4-5-F	59.67	CTG CTG CTG CTG AGA TTT ACG
Scaffold_8_mat: 66018494	PCR primer set	541 bp	1867_HemOce-sc8-3-F	59.05	GCC TCA ACC ACA ACT TCA GG
			1868_HemOce-sc8-3-R	60.03	GCC CAG CTG TTA TCA ACC CT
	Sequencing primer		1867_HemOce-sc8-3-F	-	GCC TCA ACC ACA ACT TCA GG
Scaffold_14_mat: 4777531	PCR primer set 1	560 bp	1922_HemOce-sc14-1-F	57.71	TGT CCA TGG TGA ACA GCA AT

Commented [A7]: Ashley, please check. This is different from the one in Frank's version with their comments.

			1923_HemOce-sc14-1-R	59.38	ACG TAG TTG TGG ACA GAT GCT
	Sequencing primer 1		1923_HemOce-sc14-1-R	-	ACG TAG TTG TGG ACA GAT GCT
	PCR primer set 2	507 bp	1924_HemOce-sc14-2-F	59.45	CAA GTG TGA GCA GCT TGA TTG A
			1925_HemOce-sc14-2-R	59.96	AGT TGT GGA CAG ATG CTC CAT T
	Sequencing primer 2		1925_HemOce-sc14-2-R	-	AGT TGT GGA CAG ATG CTC CAT T
Scaffold_19_mat: 62722441	PCR primer set	603 bp	1873_HemOce-sc19-3-F	60.04	TTC CTG AAG CGT GCT GCA TA
			1874_HemOce-sc19-3-R	60.18	CAT TAC ACA AAC GGT CGC GG
	Sequencing primer		1869_HemOce-sc19-1-F	-	ACT GGA CGC TTC TGC AAT CA
Scaffold_24_mat: 38442575	PCR primer set	558 bp	1877_HemOce-sc24-2-F	59.18	TGG CAA GAT CAC GGA GGT AG
			1878_HemOce-sc24-2-R	59.6	TCT ACA AAC ACT GCC CCA GG
	Sequencing primer		1877_HemOce-sc24-2-F	-	TGG CAA GAT CAC GGA GGT AG
Scaffold_28_mat: 17411576	PCR primer set + Sequencing primer set 1	281 bp	1881_HemOce-sc28-1-F	60.07	ACA GAA TTA CCC CCG GTA GG
			1882_HemOce-sc28-1-R	60.04	CTT TCC AAC CAT CCA GAG GA
	PCR primer set +	640 bp	1883_HemOce-sc28-2-F	60.18	TAC CCC CGG TAG GGA AAC AA

	Sequencing primer set 2		1884_HemOce-sc28-2-R	59.97	CAG CTC GGG TGT AGT TTG GT
	PCR primer set + Sequencing primer set 3	536 bp	1885_HemOce-sc28-3-F	59.89	AGA GGA GGG CAA AAA GGT GG
			1886_HemOce-sc28-3-R	60.11	TAG GCC ATG GGT ACC TCT CC
Scaffold_36_mat: 8326154	PCR primer set	576 bp	1889_HemOce-sc36-2-F	59.5	AAA GCA GAA AAC CTT GTG CAG T
			1890_HemOce-sc36-2-R	60.39	TGG GCC ACG ATC AAT GTC TG
	Sequencing primer		1889_HemOce-sc36-2-F	-	AAA GCA GAA AAC CTT GTG CAG T
Scaffold_51_mat: 8719242	PCR primer set	642 bp	1926_HemOce-sc51-1-F	60.88	CTC GGC GAA CAG AAC CAC AA
			1927_HemOce-sc51-1-R	59.08	TGG TCC TCT ATC TGT CGG GA
	Sequencing primer		1927_HemOce-sc51-1-R	-	TGG TCC TCT ATC TGT CGG GA
Scaffold_158_mat: 387249	PCR primer set	734 bp	1930_HemOce-sc158-1-F	59.75	GGT CAG GGA AGC TGA ACG AT
			1931_HemOce-sc158-1-R	60.18	TGA TGA AGG GCT TTT GCC CA
	Sequencing primer		1930_HemOce-sc158-1-F		GGT CAG GGA AGC TGA ACG AT

Supplementary table 3b: Primers for parents

Target candidate de novo mutation	Primer set	Fragment size	Primer name	Tm	Sequence 5' - 3'
Scaffold_4_mat: 48132425	Primer set 1	278 bp	1857_HemOce-sc4-1-F	59.96	GCA CCA CCA CCT CCA TTA CA
			1858_HemOce-sc4-1-R	57.90	TGG TCC AGT CAT GTT TGT TGT
	Primer set 2	578 bp	1859_HemOce-sc4-2-F	60.11	TTC GAA GGA CCG CAG ATG TC
			1860_HemOce-sc4-2-R	60.03	CCT CCT GCA GGG GCA TTA AA
	Primer set 3	615 bp	1861_HemOce-sc4-3-F	60.07	AAT TCG AAG GAC CGC AGA TGT
			1862_HemOce-sc4-3-R	58.86	AAG GCA GAA CAG AAG TCC CA
Scaffold_4_mat: 13524629	Primer set 1	544 bp	1918_HemOce-sc4-4-F	60.14	GGT GAA AAC GTG TTG CTG GTT
			1919_HemOce-sc4-4-R	59.38	TTC CCA CAC TGA TCC AGC TC
	Primer set 2	638 bp	1920_HemOce-sc4-5-F	59.67	CTG CTG CTG CTG AGA TTT ACG
			1921_HemOce-sc4-5-R	59.79	TCT TAT CTG TCA GCC AAG GGC
	Primer set 3	681 bp	1920_HemOce-sc4-5-F	59.67	CTG CTG CTG CTG AGA TTT ACG
			1920A_HemOce-sc4-6-R	59.0	CAG AGG CTG CTC TTG GAG TG

Scaffold_8_mat: 66018494	Primer set 1	329 bp	1863_HemOce-sc8-1-F	60.67	GGC ACG ACC ACA GCA TCT T
			1864_HemOce-sc8-1-R	60.18	CGC ATT TTA GCC TGT GGC TTT AT
	Primer set 2	532 bp	1865_HemOce-sc8-2-F	59.68	TAC TTC AGA CAC CAG GCA CG
			1866_HemOce-sc8-2-R	59.68	GCC TTC TGT CTC TTG CCT CA
	Primer set 3	541 bp	1867_HemOce-sc8-3-F	59.05	GCC TCA ACC ACA ACT TCA GG
			1868_HemOce-sc8-3-R	60.03	GCC CAG CTG TTA TCA ACC CT
Scaffold_14_mat: 4777531	Primer set 1	560 bp	1922_HemOce-sc14-1-F	57.71	TGT CCA TGG TGA ACA GCA AT
			1923_HemOce-sc14-1-R	59.38	ACG TAG TTG TGG ACA GAT GCT
	Primer set 2	507 bp	1924_HemOce-sc14-2-F	59.45	CAA GTG TGA GCA GCT TGA TTG A
			1925_HemOce-sc14-2-R	59.96	AGT TGT GGA CAG ATG CTC CAT T
	Primer set 3	596 bp	1924_HemOce-sc14-2-F	59.45	CAA GTG TGA GCA GCT TGA TTG A
			1924A_HemOce-sc 14-3-R	58.0	GAA AGA TGC TTT GAT TTA CAT GCA CC
Scaffold_19_mat: 62722441	Primer set 1	279 bp	1869_HemOce-sc19-1-F	59.96	ACT GGA CGC TTC TGC AAT CA

			1870_HemOce-sc19-1-R	59.32	GCT GAC CAC AAA AGA AGC AGT
	Primer set 2	748 bp	1871_HemOce-sc19-2-F	60.03	TGC ATG CCA CGG TCC TAT TT
			1872_HemOce-sc19-2-R	59.96	TCC ATT GAA CAC AGC CCT CC
	Primer set 3	603 bp	1873_HemOce-sc19-3-F	60.04	TTC CTG AAG CGT GCT GCA TA
			1874_HemOce-sc19-3-R	60.18	CAT TAC ACA AAC GGT CGC GG
Scaffold_24_mat: 38442575	Primer set 1	303 bp	1875_HemOce-sc24-1-F	60.09	TCT GGC TGT TCC TTT TTG AGG T
			1876_HemOce-sc24-1-R	59.96	CTT CGC CCA ACA GTT CCT CT
	Primer set 2	558 bp	1877_HemOce-sc24-2-F	59.18	TGG CAA GAT CAC GGA GGT AG
			1878_HemOce-sc24-2-R	59.60	TCT ACA AAC ACT GCC CCA GG
	Primer set 3	545 bp	1879_HemOce-sc24-3-F	59.39	CTG GTC ACA GAC CTC CAG TC
			1880_HemOce-sc24-3-R	60.25	ACT TCG CCC AAC AGT TCC TC
Scaffold_28_mat: 17411576	Primer set 1	281 bp	1881_HemOce-sc28-1-F	60.07	ACA GAA TTA CCC CCG GTA GG
			1882_HemOce-sc28-1-R	60.04	CTT TCC AAC CAT CCA GAG GA

	Primer set 2	640 bp	1883_HemOce-sc28-2-F	60.18	TAC CCC CGG TAG GGA AAC AA
			1884_HemOce-sc28-2-R	59.97	CAG CTC GGG TGT AGT TTG GT
	Primer set 3	536 bp	1885_HemOce-sc28-3-F	59.89	AGA GGA GGG CAA AAA GGT GG
			1886_HemOce-sc28-3-R	60.11	TAG GCC ATG GGT ACC TCT CC
Scaffold_36_mat: 8326154	Primer set 1	331 bp	1887_HemOce-sc36-1-F	57.94	GCA AAA GTA TGA GTT GAA TGC CT
			1888_HemOce-sc36-1-R	58.60	TTC CCT CCT CTT CCT CTC CT
	Primer set 2	576 bp	1889_HemOce-sc36-2-F	59.50	AAA GCA GAA AAC CTT GTG CAG T
			1890_HemOce-sc36-2-R	60.39	TGG GCC ACG ATC AAT GTC TG
	Primer set 3	677 bp	1891_HemOce-sc36-3-F	58.23	AGC AGA AAA CCT TGT GCA GT
			1892_HemOce-sc36-3-R	59.76	GCT CAC TAG CGT GAG GAA CA
Scaffold_51_mat: 8719242	Primer set 1	642 bp	1926_HemOce-sc51-1-F	60.88	CTC GGC GAA CAG AAC CAC AA
			1927_HemOce-sc51-1-R	59.08	TGG TCC TCT ATC TGT CGG GA
	Primer set 2	626 bp	1928_HemOce-sc51-2-F	60.33	ACA GCC CTC CAA TCT CCG TA

			1929_HemOce-sc51-2-R	59.96	AAA GGG AGC ATT CAG CGT CA
Scaffold_158_mat: 387249	Primer set 1	734 bp	1930_HemOce-sc158-1-F	59.75	GGT CAG GGA AGC TGA ACG AT
			1931_HemOce-sc158-1-R	60.18	TGA TGA AGG GCT TTT GCC CA
	Primer set 2	608 bp	1932_HemOce-sc158-2-F	58.92	TTG GTA AAG CAT TTC TGT CTC CC
			1933_HemOce-sc158-2-R	60.60	GAC CTG CTT TGC TTT TCC AGC

Point-by-point response

Reviewer #1 (Remarks to the Author):

This is a very nicely written and a really interesting piece of work: the manuscript is framed interestingly to deliver the main finding: surprisingly and intriguingly low estimate of de novo mutation rate in a shark species. This finding is further discussed sensibly, and attempts are made to pin-point explanations for the low mutation rate in sharks. This is without doubt a paper that will get people's attention. The conclusions follow from the results, and the analyses appear to be conducted rigorously. The sample size of two parents and 9 F1-generation offspring is not impressive, but probably sufficient (under the assumption that the two parents are representative of the entire species) IF the estimated mutation rate is statistically well supported. Unfortunately, no confidence intervals are provided for the μ - we believe it is critical to see these to be certain that the conclusions hold. Apart of this, we do not have any major criticism, but some suggestions and reflections that might be worth considering as follows.

Response: We agree that it is important to add confidence intervals to this estimate. We have now added this information and now report $\mu = 7 \times 10^{-10}$ (95% CI = $1.4-14.1 \times 10^{-10}$). Even the upper limit is below all other estimates listed in Table 6.

L48-49 Agreed. However, what one might be left thinking is that what role mutation rate is likely to play in restoration of genetic variation after population size bottleneck? Even if we assume that population experiences serve bottleneck and loses much of its genetic variation, many other things apart from mutation rate can be more important for the recovery of the genetic variability. These include for instance longevity/age structure – long-lived species like sharks can buffer against loss of diversity better than short-lived species. Other important factor is metapopulation structure – immigrants from other subpopulations can help to restore the lost diversity much faster than any increase in mutation rate. In this perspective, one could argue that the argument put forth on these lines is in principle correct, but at the same time, not necessarily very relevant unless one makes lots of assumptions. Some readers might go even so far as to claim that this argument is a red herring.

Response: We agree with the reviewer that how a bottleneck will affect genetic diversity depends on the length of the bottleneck and if it affects the entire species or just local populations. This is the reason why we write “a sustained population bottleneck” and “may be hampered”. We therefore think that this is a relevant comment to our results.

L54. “adaptive potential” rather than “adaptive ability”?

Response: Done

L172. See above. There should be a way to estimate confidence intervals for this estimate.

Response: We have included the confidence interval

L181. The callable genome (CG) size applied here seemed quite close to the total genome size (more than 3Gb) rather than what had been mentioned e. arlier ranging from 8% to 11% of the reference genome size. Could it be that this refers to a sum of CGs for all offspring that were used to calculate an overall μ for the entire pedigree in this paper? μ for a

pedigree is normally calculated by averaging the per-offspring μ , rather than summing over them. Since the estimate of μ depends heavily on the size of CG, it is quite important that the authors disclose (perhaps in the supplementary materials) how exactly this key parameter was calculated. Also, we note that the CG ranged from 45% to 91.5% in the earlier studies included into the “Mutationathon” paper in eLife. Hence, it is a bit worrying to see very small CG estimates (8 to 11%) in this paper. Some discussion about this in the ms could be in place. How do the authors explain the small callable genome size despite the deep sequencing coverage of the samples?

Response: Here the callable genome (CG) refers to the sum of CGs for all offspring. We have clarified this in the text: “We estimated the mutation rate per site per generation by dividing the expected number of true de novo mutations identified by 2 x the total number of callable sites screened across the pedigree”. The callable fraction (~10%) is not massively lower than that for Common Carp (17% in Bergeron et al). The procedure we use, in particular with mappability 1 being a criterion, is quite conservative and sensitive to repeat content, which is high in this genome (44%). The callable fraction does not bias the rate in any particular direction, since it restricts both mutated sites and non-mutated ones.

L186. Perhaps make it clear that this is the long-term equilibrium N_e – not the current.

Response: Agreed and changed.

L203. Generalization from one shark species (two parents) to entire shark clade is quite a jump. In the same vein, isn't this particular species quite exceptional in its biology? It lives in shallow waters and can tolerate extreme hypoxia and even anoxia. Is it thinkable that these stressful conditions have selected for efficient DNA reparation machinery and the μ for this species might not be representative for the entire shark clade?

Response: We are not aware of anything exceptional about Epaulette sharks among sharks in general. Estimated branch lengths on phylogenies (based on mitochondrial data) suggest their substitution rate is typical among sharks (see Naylor et al., Elasmobranch phylogeny, Biology of Sharks, 2012, chapter2, see tree on page 41). Indeed deep water squaloid sharks have slower rates of evolution than do Epaulette sharks. This may imply that squaloid sharks will show an even lower mutation rate.

L246-255. See the earlier comment above: this seems bit naïve conclusion given that other factors than mutation rate are likely to be more important for restoring genetic variability. Also, it would be perhaps a good idea discuss the interpretation of the N_e estimate and its assumptions.

Response: We have modified the text and clarified that this estimate refers to long-term effective population size and that the loss of genetic diversity is most severe after a sustained bottleneck and if it has affected the entire species population, which means that genetic diversity cannot be restored by gene flow.

L260-262. Please provide reference.

Response: A reference has been added.

L268. The parental fish were kept in captivity for a very long time. Is thinkable that the meioses where the mutations occurred took place during the captive period? Is it thinkable that conditions in aquarium (e.g. temperature) might have influenced the mutation rates? Should more details be given on the rearing conditions?

Response: While the reviewer is correct that the parental fish were kept in captivity for a considerable period (greater than ten months) prior to collection of offspring, it is perhaps important to note in the context of the reviewer's comments, that the parental fish were caught as sexually mature adults and thus had spent the majority of the lifespan in the wild. While information on the development and timing of germ cell formation is scant for cartilaginous fish, and to our knowledge not evident in the literature for epaulette sharks, it has been proposed that initiation of oogenesis is an early event in cartilaginous fish (McMillin, Ovarian follicles. Pages 67–208 in D. B. McMilan: Fish histology: female reproductive system. 2007; Prisco et al, Reviews in Fish Biology and Fisheries 17:1-10, 2007), suggesting that at least some meiosis may have occurred in the wild for these animals.

Housing conditions (physical parameters) were maintained as close as possible to natural conditions. We, however, do agree with the reviewer that more details on the housing conditions would strengthen the submission. We have added additional detail on husbandry conditions in the methods, including descriptions of temperature, photoperiod, and feeding regime. We also note that water temperature was maintained at approximately 25°C, in line with previous studies of Epaulette captive breeding (West and Carter, Observations on the development and growth of the epaulette shark *Hemiscyllium ocellatum* (Bonnaterre) in captivity. Journal of Aquariculture & Aquatic Sciences 5:111-117, 1990) and within the range of seasonal water temperatures along the Great Barrier Reef (Wheeler et al, Future thermal regimes for epaulette sharks (*Hemiscyllium ocellatum*): growth and metabolic performance cease to be optimal. *Sci Rep* 11, 454, 2021). Therefore, we do not expect our calculated mutation rate to be different from Epaulette sharks in their native distribution.

L453. From what was described here, the CG had been worked out by applying the “in-house filtering pipeline” which was a series of filters to ensure that the parental homozygotes and offspring heterozygotes are both true. It was nice to see that the filtering bounds were set based on distributions. However, Mendelian violation should be one of the steps identifying de novo mutations (numerator) rather than defining CG (denominator). It is not quite clear if this is how the authors' pipeline treated these.

Response: In our pipeline the high-confidence heterozygous sites were used solely to define the filtering cut-offs applied to the whole dataset. Sites that passed the filtering criteria represented the “callable” fraction of the genome in each trio that could be confidently screened for mendelian violations. Within the callable fraction of the genome, all candidate mutations were identified on the basis that they violated mendelian expectations i.e. an offspring carried an allele that was absent in the parents.

L460. Normally there is also an upper bound for allelic balance (AB) filter preventing offspring genotypes to be homozygotes for the alternative allele (1/1).

Response: It is not clear which line in the manuscript the reviewer is commenting on. If this comment is not taken care off through our revision of the methods description, please advise.

Fig 6. We believe standard errors or confidence intervals should be shown in this figure. Without seeing them, there is no way to know what to think of the data in the figure.

Response: We are not able to calculate confidence intervals for the data from previous studies because we do not have access to the data required to make these calculations. However, we have added a 95% CI for the epaulette shark to the text and show that the upper limit of this CI is below the estimated mean for all other species listed in Fig. 6.

In addition to all the comments above, we note that there are number of small technical glitches in the text (missing full-stops, etc).

Response: We have carefully checked the revised manuscript and corrected typos and errors.

Reviewer #2 (Remarks to the Author):

This is a very well written manuscript and I enjoyed reading it. I have made comments and suggestion directly on the Word .doc.

Response: We thank the reviewer for careful reading. All typos and errors have been corrected.

One of my concerns with the paper is the estimation of the mutation rate--please see comments in the Word .doc.

Response: Please, see below.

Commented [A1]: Up to 96 in offspring and up to 135 for the sire

Response: We have updated the text as follows: “As identification of de novo mutations requires high sequence coverage, we sequenced each offspring to ~49-82 x coverage (Supplementary Table 2). The resulting sequences, along with parental Illumina sequences (~113-135 x coverage) previously generated for genome assembly construction ...”

Commented [A2]: But to dismiss these two occurrences changes the mutation rate, and instead of $(4.8 / (2 \times 3,691,810,944) = 6.5 \times 10^{-10})$, we would have $(6.8 / (2 \times 3,691,810,944) = 9.21 \times 10^{-10})$.

Response: Since our initial submission we have been able to dismiss a further candidate mutation (scaffold_4_mat, position 13524629) via plasmid cloning. Whole-plasmid ONT sequencing of four clones provided no support for the alternative genotype in ind2048. This left a single candidate mutation (scaffold_28_mat, position 17411576) as un-validated. In line with the reviewer’s concerns we now take a conservative approach when estimating the mutation rate by including the unvalidated candidate mutation in our estimate (rather than extrapolating the expected number of true mutations using the false positive rate). The mutation rate is now calculated as: $5 / (2 \times 3,691,810,944) = 6.8 \times 10^{-10}$. By correcting for the estimated false negative rate we obtain: 7×10^{-10} .

Commented [A3]: The calculation of the mutation rate needs to be explained better. Why not use EVERY possible mutation (see below)?

Response: Please, see below.

Commented [A5]: The way this is written, it seems like the authors look at mutations like the following:

Mom CAAT/CAAT

Dad CAAT/CAAT

Offspring: CAAT/CAAA

Why just these mutations? Why not looking at others, e.g.:

Mom CAAT/CAAT

Dad CAAA/CAAA

Offspring: CAAT/CAAG or CAAA/CAAG

It seems like you will miss mutations and underestimate the mutation rate. This is obviously at the heart of the mutation rate calculation and a better justification is needed regarding why only certain mutations were considered.

Response: The reason why these are not included is that the sites where the parents are homozygous for opposite alleles are so few (<0.1%) compared with those where parents are homozygous for the same allele. Thus, including these sites will not affect the estimate. The number of callable sites (where both parents are homozygous for the same allele) was on average about 400 Mb in this study while the number where parents are homozygous for opposite alleles are only 281,143 bp and we detected no de novo mutations at these sites in this dataset. We prefer not to include this information in the paper because it will not change our estimate and to the best of our knowledge these sites have not been used in previous studies on pedigree-based estimates of mutation rate. In fact, we think these sites are more unreliable because they probably are overrepresented for errors, for instance if there are duplicated sequences in the genome that are not properly annotated.

REVIEWER COMMENTS

Reviewer #1 (Remarks to the Author):

We appreciate the effort authors have done to revise the manuscript, but we are still left with some concerns.

First, it is nice to see that confidence intervals for DNM rate are now provided. However, I was unable to see how these estimates were obtained (i.e. the description of the procedure to obtain μ is not described in the methods unless I have overlooked something). A more serious concern is that the provided upper CI overlaps largely for instance with the lower CI for the DNM rate estimate for the Atlantic herring (1.8×10^{-10} , CI: $11 - 27 \times 10^{-10}$). What this means is that although the point estimate for Epaulette shark is low, it is not significantly lower than for Atlantic herring. What this means is that there is no evidence to claim that DNM rate estimate for this shark species is exceptionally low. Authors' response that "Even the upper limit is below all other estimates listed in Table 6" is misleading because it is the confidence intervals, rather than point estimates, which are relevant in this context (also, there is no Table 6 in the ms). Based on this, one could say that the title could be perceived as misleading claiming extremely low de novo mutation rate in the epaulette shark (more accurate would be "Low de novo mutation rate in the epaulette shark").

In response to comment regarding L203 the authors responded that they are not aware of anything exceptional about Epaulette sharks among sharks in general. We wrote earlier that "It lives in shallow waters and can tolerate extreme hypoxia and even anoxia. Is it thinkable that these stressful conditions have selected for efficient DNA repair machinery and the μ for this species might not be representative for the entire shark clade?" Do other sharks generally tolerate extreme hypoxia and anoxia? These are significant physiological features that do seem to differentiate Epaulette sharks from other sharks and potentially explain their low looking mutation rates?

In response to query regarding confidence intervals in Figure 6, it is hard to understand why the authors would need the data to obtain them. The confidence intervals are readily available from relevant publications (e.g. the Atlantic herring estimates referred to above). For

comprehending what the figure really shows, it is critical to insert confidence intervals in it. Perhaps unfortunately, this will reveal that there is a lot of uncertainty around the point estimates and the Epaulette shark estimate is not that different from e.g. Atlantic herring estimate.

Finally, what was overlooked during the previous reading is that the authors have used a genotypic quality threshold of < 20 . This is quite a generous value as most (but not all) studies use a higher threshold (e.g. < 60 , < 80 or even higher). What this means is that the risk of calling false positives is increased with the low threshold value. One is left to wonder what would have happened if the authors had increased the threshold? Would they have not detected any mutations at all?

Reviewer #2 (Remarks to the Author):

The reviewers have done a good job addressing my first round of comments. I only have a few additional items to be addressed.

Line 431: <https://www.uniprot.org/statistics/Swiss-Prot> appears to be a broken link

Line 603: https://genomeark.github.io/genomeark-all/Hemiscyllium_ocellatum/ also appears to be a broken link

Lines 726-728: ref 53--italicize species name and add spaces where needed

I have no further comments. Well done!

Point-by-point response

Reviewer #1 (Remarks to the Author):

We appreciate the effort authors have done to revise the manuscript, but we are still left with some concerns.

First, it is nice to see that confidence intervals for DNM rate are now provided. However, I was unable to see how these estimates were obtained (i.e. the description of the procedure to obtain CIs is not described in the methods unless I have overlooked something). A more serious concern is that the provided upper CI overlaps largely for instance with the lower CI for the DNM rate estimate for the Atlantic herring (1.8×10^{-10} , CI: $11-27 \times 10^{-10}$). What this means is that although the point estimate for Epaulette shark is low, it is not significantly lower than for Atlantic herring. What this means is that there is no evidence to claim that DNM rate estimate for this shark species is exceptionally low. Authors' response that "Even the upper limit is below all other estimates listed in Table 6" is misleading because it is the confidence intervals, rather than point estimates, which are relevant in this context (also, there is no Table 6 in the ms). Based on this, one could say that the title could be perceived as misleading claiming extremely low de novo mutation rate in the epaulette shark (more accurate would be "Low de novo mutation rate in the epaulette shark").

Response: Details of how CIs were obtained are now reported in the methods - see section "Estimation of 95% confidence interval and statistical significance". We have now tested whether our estimated mutation rate for the epaulette shark differs significantly from the mutation rate estimated for the Atlantic herring, details of which have been added to the text. In short, although there is minor overlap between the two CIs, we are now able to state that the mutation rate estimated for the epaulette shark is indeed significantly lower than that of the Atlantic herring ($P=0.0009$), which upholds our conclusion that this species has the lowest mutation rate directly investigated in a vertebrate to date.

In response to comment regarding L203 the authors responded that they are not aware of anything exceptional about Epaulette sharks among sharks in general. We wrote earlier that "It lives in shallow waters and can tolerate extreme hypoxia and even anoxia. Is it thinkable that these stressful conditions have selected for efficient DNA repair machinery and the μ for this species might not be representative for the entire shark clade?" Do other sharks generally tolerate extreme hypoxia and anoxia? These are significant physiological features that do seem to differentiate Epaulette sharks from other sharks and potentially explain their low looking mutation rates?

Response: Yes, other sharks also tolerate extreme hypoxia and anoxia. While it is clear that the epaulette shark is the most studied species (probably due to ease of study and captive access) other species clearly show pro-survival adaptive response to hypoxic conditions (Bouyoucos et al. J. Exp. Biol. 223: 1, 2020; Butler & Taylor J. Exp. Biol. 63: 117, 1975; Svenson et al. J. Exp. Zool. 303: 154; 2005; Devaux et al. J. Exp. Biol. 222: 1, 2019; Piiper et al. Comp. Biochem. Physiol. 36:513, 1970).

With respect to the concern that stressful conditions have selected in the Epaulette shark for a more efficient DNA repair machinery and thus the μ which we found for this species might be not representative, we did a positive selection analysis. We downloaded the annotated gene sets of all high quality shark reference genomes on NCBI (great white shark, whale shark, nurse shark, shortfin mako shark, ocean whitelip shark, lemon shark, and elephant shark as outgroup). These species represent a broad spectrum of life history traits. With NCBI-annotated protein sequences of great white shark as the queries, we retrieved the orthologous sequences in the other six sharks using Exonerate. After filtering, the one-to-one orthologs were used for sequence alignment using MAFFT

and PAL2NAL, phylogeny reconstruction using RAxML; and positively selected gene analysis using codeml in branchsite model. Then we asked which genes are under positive selection specifically in the Epaulette shark. We retrieved 90 genes with a clear signal of selection (see attachment). We checked for enrichment and known functions using the Gene Ontology Resource. None of the genes is listed under GO term “DNA repair (GO:0006281), (577 genes, 1539 annotations)”.

In response to query regarding confidence intervals in Figure 6, it is hard to understand why the authors would need the data obtain them. The confidence intervals are readily available from relevant publications (e.g. the Atlantic herring estimates referred above). For comprehending what the figure really 6 shows, it is critical to insert confidence intervals in it. Perhaps unfortunately, this will reveal that there is lot of uncertainty around the point estimates and the Epaulette shark estimate is not that different from e.g. Atlantic herring estimate.

Response: We have added 95% confidence intervals (CIs) for other vertebrates to supplementary table 1, these are not included the Figure 8 as CIs are not reported in the literature for many of the species concerned. Of those where CIs are reported, only the Atlantic herring had minor overlap with our estimated 95% CI for the Epaulette shark. Despite this overlap we have determined that herring and epaulette shark point estimates differ significantly (see response to comment #1).

Finally, what was overlooked during the previous reading is that the authors have used genotypic quality threshold of < 20. This is quite generous value as most (but not all) studies use higher threshold (e.g. < 60, < 80 or even higher). What this means is that the risk of calling false positives is increased with the low threshold value. One is left to wonder what would have happened if authors have increased the threshold? Would they have not detected any mutations at all?

Response: While there is an increased risk of calling false positive candidate mutations at genotype quality threshold 20, all candidate mutations identified in this study have been validated by Sanger sequencing (with the exception of the candidate mutation on scaffold_28_mat). Where mutations are confirmed by Sanger Sequencing (which has over 99% accuracy) we can be confident that they are true mutations. In fact, mean GQ at candidate mutations was >80, with 9/12 candidates having genotype quality scores of 99 (accuracy > 99.9999%).

Reviewer #2 (Remarks to the Author):

The reviewers have done a good job addressing my first round of comments. I only have a few additional items to be addressed.

Line 431: <https://www.uniprot.org/statistics/Swiss-Prot> appears to be a broken link

Response: Broken link has been fixed.

Line 603: https://genomeark.github.io/genomeark-all/Hemiscyllium_ocellatum/ also appears to be a broken link

Response: Broken link has been fixed.

Lines 726-728: ref 53--italicize species name and add spaces where needed.

Response: Spaces added and species name italicized.

REVIEWERS' COMMENTS

Reviewer #1 (Remarks to the Author):

I think the authors have have addressed most of my concerns in satisfactory manner albeit I still think showing the confidence intervals in fig 8 for the species for which they are available would have been really informative and transparent reporting.

One minor thing: the estimate for the Atlantic herring in Fig 8 and Supplementary table 1 is according to Feng et al 2017 (their p7.) $18 * 10^{-10}$, and not $20 * 10^{-10}$ as reported in this ms. I know that in their abstract they reported 20 for reasons unknown to me, but I assume the estimate reported in the main text is what seems to most accurate one.

Point-by-point response

Reviewer #1 (Remarks to the Author):

I think the authors have addressed most of my concerns in satisfactory manner albeit I still think showing the confidence intervals in fig 8 for the species for which they are available would have been really informative and transparent reporting.

Response: We have already responded to this point with the second revision. We agree that it would have been informative for the reader to show the confidence intervals (CIs) in fig 8 only for a subset of species the respective publications give a CI. We have added 95% CIs for other vertebrates to supplementary table 1. We decided not to include CIs in Figure 8 as CIs are not reported in the literature for many of the species concerned and therefore the figure might look confusing to the non-specialist reader.

One minor thing: the estimate for the Atlantic herring in Fig 8 and Supplementary table 1 is according to Feng et al 2017 (their p7.) 18×10^{-10} , and not 20×10^{-10} as reported in this ms. I know that in their abstract they reported 20 for reasons unknown to me, but I assume the estimate reported in the main text is what seems to most accurate one.

Response: The estimate of 20×10^{-10} is what Feng et al calculated when correcting for the false negative rate. In our manuscript on the mutation rate of the Epaulette shark we also report the false negative corrected rate as the final estimate. We are confident that false negative correction will provide a more accurate estimate.